# Learning Task-Agnostic Motifs to Capture the Continuous Nature of Animal Behavior

## Abstract

Animals flexibly recombine a finite set of core motor motifs to meet diverse task demands, but existing behavior segmentation methods oversimplify this process by imposing discrete syllables under restrictive generative assumptions. To better capture the continuous structure of behavior generation, we introduce motif-based continuous dynamics (MCD) discovery, a framework that (1) uncovers interpretable motif sets as latent basis functions of behavior by leveraging representations of behavioral transition structure, and (2) models behavioral dynamics as continuously evolving mixtures of these motifs. We validate MCD on a multi-task gridworld, a labyrinth navigation task, and freely moving animal behavior. Across settings, it identifies reusable motif components, captures continuous compositional dynamics, and generates realistic trajectories beyond the capabilities of traditional discrete segmentation models. By providing a generative account of how complex animal behaviors emerge from dynamic combinations of fundamental motor motifs, our approach advances the quantitative study of natural behavior.

## 1 Introduction

A critical direction in animal behavior research has been identifying recurring patterns, often referred to as stereotyped behavioral syllables, like back grooming, running, and sniffing, directly from large-scale behavior recordings. Behavior segmentation methods (Wiltschko et al., 2015; Weinreb et al., 2024; Luxem et al., 2022; Hsu & Yttri, 2021; Berman et al., 2014) seek to uncover such structured patterns in behavior by dividing continuous pose trajectories into discrete syllables. Classic behavior segmentation approaches can be categorized into three groups: **(1)** supervised classification (Marks et al., 2022; Segalin et al., 2021), **(2)** clustering-based methods (Hsu & Yttri, 2021; Berman et al., 2014; Whiteway et al., 2021), and **(3)** hidden-Markov-model(HMM) based methods (Wiltschko et al., 2015; Weinreb et al., 2024; Luxem et al., 2022; Costacurta et al., 2022). The segmented syllables can then be used to build structured representations of movement for downstream neurobehavioral study.

However, existing behavior segmentation methods (Wiltschko et al., 2015; Weinreb et al., 2024; Luxem et al., 2022; Hsu & Yttri, 2021; Berman et al., 2014) overlook several features of the behavior data. First, **continuity**, they model continuous behavior as combinations of discrete action syllables, which oversimplifies the inherently continuous nature of movement and introduces ambiguity during action transitions. Therefore, they may fail to capture the details of delicate behavior dynamics. Second, **compositionality**, they often extract complex coordinated body movements as abstract syllables that fail to capture how individual body parts contribute to different motions. Therefore, they fail to reveal the connections and distinctions between syllables. For example, back and side grooming both involve similar forelimb movements combined with different turning dynamics, and sniffing may occur while walking or sitting, with similar head motion but distinct lower-body patterns. Third, **long-term dependency**. They either ignore temporal dependency between actions or only consider a very short time window when encoding the behavior. Therefore, they often fail to capture the long-term and multi-scale property of syllables (See Appendix. I for discussions on temporal dependency). Apart from these explicit features, most models are either non-generative (e.g., clustering Hsu & Yttri (2021); Berman et al. (2014)) or rely on restrictive generative assumptions (e.g., linear dynamics and Markov models Wiltschko et al. (2015); Weinreb et al. (2024); Luxem et al. (2022)), often leading to unrealistic synthesized behaviors.

To address these limitations, we introduce a new perspective: modeling behavior under the reinforcement learning (RL) framework. We study behavioral dynamics and motor motifs by inferring the animal's policy through an RL-based imitation learning (IL) framework. It not only enables more

realistic behavior generation through RL but also allows us to discover reusable motor motif sets to construct a policy driven by internal rewards. By viewing behavior through this lens, we gain a more flexible, generative, and interpretable understanding of motor motifs that go beyond the constraints of discrete segmentation. Note that Aldarondo et al. (2024) also applied RL-based IL to analyze animal behavior, but without parsing long untrimmed behaviors into fine-grained, interpretable motor motifs, so their work lies outside the scope of behavior segmentation considered here.

We hypothesize that animals draw from a fixed set of core motor motifs to construct diverse movements over long behavioral trajectories (Santuz et al., 2019; Flash & Hochner, 2005). Building on this, we propose *Motif-based Continuous Dynamics discovery (MCD)* to parse long trajectories and uncover motifs and policies that reflect behavioral dynamics. Concretely, we learn interpretable latent representations, or **motif sets**, via spectral decomposition–based representation learning in RL (Dai et al., 2014; Ren et al., 2023; Shribak et al., 2024). These motifs correspond to low-level motion patterns serving as modular building blocks of behavior and can involve different body parts. For instance, face grooming (forepaw-to-face) and body grooming (torso strokes) share grooming motifs while engaging distinct body parts. These motifs can then be used to sufficiently represent policies that characterize complex high-level behaviors. Finally, we apply imitation learning to train motif-based policies from demonstrations. This framework leverages RL in two aspects: (1) motifs are inferred through RL-based representation learning, and (2) we use policies formed from motifs to characterize behavioral dynamics. As shown later, both aspects avoid any model assumptions while capturing motifs and policies that faithfully reflect behavioral dynamics.

Another key innovation in constructing policies from motifs is that each motif's contribution evolves continuously over time, reflecting dynamic behavioral changes. Some motifs may be brief, others prolonged, and multiple motifs can be active simultaneously to build ongoing movement (Fig. 1). This flexible, compositional view cannot be achieved with traditional discrete syllables. MCD thus provides a nuanced account of how complex actions arise from dynamic

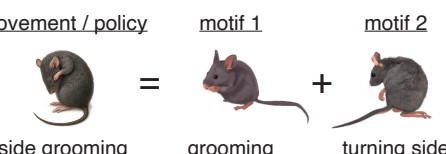

Figure 1: A policy for a movement can be seen as a blend of "vocabularies" from a dictionary containing fundamental motor motifs.

motif combinations and enables testing whether fine-grained motifs depend on specific neural circuits. Compared with prior segmentation methods, MCD offers *soft segmentation* that captures continuous time-varying processes rather than discrete switches. Further, theoretically the horizon in RL is infinite, enabling longer temporal dependency modeling.

Note that there is a rich literature in robotics on learning skills through imitation learning (Paraschos et al., 2013; Lioutikov et al., 2017; Li et al., 2017; Ajay et al., 2021; Peng et al., 2022; Kuang et al., 2025). We use RL-based imitation not to compete with existing skill learning methods, as most are not suited to the behavior segmentation task for neuroscience study in this paper. Rather, our goal is to employ the RL framework as a principled way to characterize the continuous nature of animal behavior while rendering fine-grained motor motifs, analogous to how *Keypoint-MoSeq* relies on an SLDS framework (Weinreb et al., 2024) and *VAME* relies on an autoencoder framework (Luxem et al., 2022). To summarize, we contribute to *behavior understanding for neuroscience and neuroethology* through the following points:

- We introduce the first RL-based IL framework for behavior segmentation, a fundamental advance since RL naturally treats behavior as a decision-making process shaped by policies and rewards. Unlike dynamics-based methods, it explains why behaviors occur, not just how they unfold.
- Within this framework, we propose RL-based representation learning to discover motif-based policies. The learned motifs and policies do not rely on dynamics assumptions or any model assumptions. They can faithfully characterize behavioral dynamics without the mismatch issues of prior segmentation methods.
- Our method reveals the continuous, compositional and long-term dependent nature of animal behavior, providing a nuanced understanding of how complex behaviors emerge from dynamic combinations of fundamental motor motifs. It moves beyond the discrete segmentation assumptions of existing methods.

## 2 PRELIMINARIES

**Markov Decision Processes (MDP).** To define motifs that characterize animal behaviors, we begin by modeling the observed behavioral trajectories within the framework of MDPs. Formally, an MDP is defined as a tuple $\mathcal{M} = (\mathcal{S}, \mathcal{A}, r, P, \rho, \gamma, H)$, where $\mathcal{S}$ denotes the state space, capturing both the

environment and an animal's condition—for example, positions of pose keypoints; $\mathcal{A}$ is the action space denoting feasible movements; $r : \mathcal{S} \times \mathcal{A} \to [0, 1]$ is a reward function encoding the immediate utility toward an internal goal; $P : \mathcal{S} \times \mathcal{A} \to \Delta(\mathcal{S})$ is the transition operator, with $\Delta(\mathcal{S})$ representing distributions over $\mathcal{S}$; $\rho \in \Delta(\mathcal{S})$ is the initial state distribution; $\gamma \in (0, 1)$ is a discount factor; and $H$ is the time horizon. A policy $\pi : \mathcal{S} \times [H] \to \Delta(\mathcal{A})$ is a conditional distribution over actions given a state for each time-step. We assume that an animal generates pose trajectories by following such an MDP where the reward function reflects an intrinsic motivation driving behavior. The behavior is governed by a policy that seeks to maximize this internal reward.

Following standard notations, we define the value function $V(s) := \mathbb{E}\left[\sum_{t=0}^{H} \gamma^t r(s_t, a_t) | s_0 = s\right]$ and the action-value function $Q(s, a) = \mathbb{E}\left[\sum_{t=0}^{H} \gamma^t r(s_t, a_t) | s_0 = s, a_0 = a\right]$, which are the expected discounted cumulative rewards when executing policy $\pi$. From the above definition, we can establish the following Bellman relationship:

$$Q_h(s, a) = r(s, a) + \gamma \mathbb{E}_{s' \sim P(\cdot|s,a)}\left[V_{h+1}(s')\right], \quad V_h(s) = \mathbb{E}_{a \sim \pi(\cdot|s)}\left[Q_h(s, a)\right]. \tag{1}$$

**Offline Imitation Learning.** We use imitation learning(IL) to find a policy $\pi$ that mimics animal behavior. In the offline IL setting, we cannot interact with the MDP environment to collect samples using policy $\pi$, but can only access a dataset of transitions sampled from the MDP by the expert, $\mathcal{D} = \{(s_i, a_i, s_i') | (s, a) \sim \tau^e, s' \sim P(\cdot|s,a), i = 1, 2, ..., N\}$, where $\tau^e$ is the data distribution of state-action pairs generated by the expert which is the animal in this study.

## 3 MOTIF-BASED CONTINUOUS DYNAMICS (MCD) DISCOVERY

Given the MDP definition, we frame motif discovery from a control-theoretic perspective. In this view, motifs are the fundamental components that enable the construction of diverse policies and reward functions, and thus help explain the motivation behind observed behaviors.

**Definition 1** (Motif Set). *Given an arbitrary transition kernel $P(\cdot|s, a)$ in an MDP, we can express it via a spectral decomposition:* $\quad P(s'|s, a) = \phi(s, a)^\top \mu(s')q(s'),$ (2) *where $\phi : \mathcal{S} \times \mathcal{A} \to \mathbb{R}^d$, $\mu : \mathcal{S} \to \mathbb{R}^d$, and $q \in \Delta(\mathcal{S})$ is a parametrized probability distribution over the state space. We define the function $\phi$ as the **motif set**. The reward function is then parametrized linearly as $r(s, a) = \phi(s, a)^\top w$.*

Spectral decomposition has been widely studied in RL representation learning (Ren et al., 2023; Zhang et al., 2022; Shribak et al., 2024). We adopt this approach here to define motifs given the transition kernel and define rewards accordingly. Since spectral decomposition derives latents directly from the transition kernel without model assumptions, motif learning is thus independent of model assumptions and faithfully reflects the motifs present in the behavior data.

Given the motif definition, substituting Eq. 2 and the linearized reward model into the Bellman equation (Eq. 1), we get:

$$Q(s, a) = r(s, a) + \gamma \int V(s')P(s'|s, a)\mathrm{d}s' = \phi(s, a)^\top\left[w + \gamma \int V(s')\mu(s')q(s')\mathrm{d}s'\right] = \phi(s, a)^\top u, \tag{3}$$

where $u = w + \gamma \int V(s')\,\mu(s')\,q(s')\,\mathrm{d}s'$. Thus, the action-value function $Q(s, a)$ can be expressed as a linear combination of motif features $\phi(s, a)$, offering a convenient way to link motifs to the policy. Following the maximum entropy reinforcement learning framework (Haarnoja et al., 2018), we assume the animal's objective is to maximize the expected reward augmented by the policy's entropy. Under this assumption, the optimal max-entropy policy $\pi(a|s)$ can be shown to follow:

$$\pi(a|s) = \arg\max_\pi \ [\mathbb{E}_\pi[Q(s, a)] + H(\pi)] = \frac{\exp(\phi(s, a)^\top u)}{\sum_{a' \in \mathcal{A}} \exp(\phi(s, a')^\top u)}, \tag{4}$$

where $H(\pi) := \sum_{a \in \mathcal{A}} \pi(a|s) \log(\pi(a|s))$ is the entropy.

**Proposition 1.** *The policy in Eq. 4 is not based on any model assumption but emerges naturally as the max-entropy policy based on spectral decomposition of the transition kernel. Furthermore, the learned motifs $\phi$ can represent any max-entropy policy through an appropriate choice of $u$.*

The reason we define $\phi(s, a)$ as *motifs* is that $\phi$ provides the linear basis for the environment transition, policies, and rewards. Policies characterize behavioral dynamics by describing action tendencies conditioned on state (*how behaviors evolve*). Combined with the environment dynamics $P(s'|s, a)$, it induces the transition distribution $P(s'|s) = \sum_a P(s'|s, a)\,\pi(a|s)$, which governs the evolution of behavior trajectories, as is often directly modeled in dynamics-based methods (Wiltschko et al., 2015;

Weinreb et al., 2024). Rewards, in turn, reflect the underlying driving factors of these trajectories (*why behaviors evolve*). Moreover, because $\phi$ is derived solely from the transition dynamics $P(s'|s, a)$, it remains independent of any specific reward or task. In this sense, $\phi(s, a)$ encodes intrinsic, general-purpose motor motifs available to animals, while the weight vector $u$ captures task-specific modulations required to produce behavior aligned with different goals. Thus, we can interpret behavioral trajectories through the lens of motifs $\phi$.

From Def. 1 and Prop. 1, we conclude that the learned motifs and policies do not rely on model assumptions, yet the policies faithfully capture behavioral dynamics as action tendencies conditioned on state. Thus, our method is assumption-free while capturing dynamics, unlike classification/clustering methods (no dynamics) or dynamics-based methods (restrictive assumptions).

Next, we introduce how to learn $\phi(s, a)$ and $\mu(s')$ (motif discovery), as well as $u$ (motif-based policy learning that characterizes the continuous behavioral dynamics) from demonstrations. The learning procedure differs depending on the nature of the behavior data (i.e. discrete or continuous).

### 3.1 DISCRETE VERSION

**Motif discovery.** For discrete state-action spaces, we apply spectral methods such as singular value decomposition (SVD) (Golub & Reinsch, 1971; Golub & Van Loan, 2013; Trefethen & Bau, 2022) or spectral decomposition representation (Ren et al., 2023; HaoChen et al., 2021) to learn the representations $\phi(s, a), \mu(s') = \arg\min_{\phi, \mu} ||P(s'|s, a) - \phi(s, a)^\top \mu(s') q(s')||^2$. The resulting motif set $\phi(s, a)$ is then used in the subsequent policy learning stage.

**Motif-based policy learning.** We now learn the policy $\pi(a|s)$, parameterized by Eq. 4, using maximum likelihood estimation (MLE), i.e., by optimizing the following objective to solve for $u$:

$$\max_u \ \mathbb{E}_{(s,a) \sim \tau^e} \left[ \log \pi(a|s) \right] = \max_u \ \mathbb{E}_{(s,a) \sim \tau^e} \left[ \log \frac{\exp(\phi(s, a)^\top u)}{\sum_{a' \in \mathcal{A}} \exp\left( \phi(s, a')^\top u \right)} \right]. \tag{5}$$

### 3.2 CONTINUOUS VERSION

While learning from discrete data is relatively straightforward, the continuous case presents additional challenges for two main reasons. First, in the motif discovery step (Eq. 2), the decomposition $P(s'|s, a) = \phi(s, a)^\top \mu(s') q(s')$ is too restrictive to capture the complexity of continuous behavioral dynamics, such as pose transitions in freely moving animals (Weinreb et al., 2024). For example, consider a common and biologically plausible behavioral model: $s' = h(s, a) + \epsilon$, where $h$ is a dynamics function and $\epsilon$ is Gaussian noise. This additive structure, widely used in behavioral modeling, contrasts with the multiplicative form $\phi(s, a)^\top \mu(s') q(s')$, suggesting that parameterizations preserving additive relationships between $\{s, a\}$ and $s'$ are more suitable. Second, in motif-based policy learning, for discrete datasets with a small action space, the denominator (partition function) in Eq. 5 is easy to compute. But for continuous data, the action space is infinite, making it infeasible to enumerate all actions and integrate. In light of these challenges, we adopt an alternative approach to learn the motif representations and policy in continuous state-action spaces.

**Motif discovery.** We model $P(s'|s, a)$ as an energy-based model (EBM) (Shribak et al., 2024):

$$P(s'|s, a) = q(s') \exp\left( \psi(s, a)^\top \nu(s') - \log Z(s, a) \right), Z(s, a) = \int q(s') \exp(\psi(s, a)^\top \nu(s')) ds', \tag{6}$$

where $\psi : \mathcal{S} \times \mathcal{A} \to \mathbb{R}^g$ and $\nu : \mathcal{S} \to \mathbb{R}^g$ are neural-network feature maps. Here, $Z(s, a)$ is an intractable partition function. Compared to the unnormalized inner-product model (Eq. 2), this EBM formulation yields smooth, normalized probabilities and stable gradients, leading to more effective and generalizable motif representations.

**Proposition 2** (Connection to Motif Definition). *Given the EBM model in Eq. 6, the transition kernel can be approximated by $P(s'|s, a) \approx \phi(s, a)^\top \mu(s') q(s')$, where $\phi(s, a) \in \mathbb{R}^d$ is an explicit function of $\psi(s, a)$ and the partition function $Z(s, a)$, and $\mu(s') \in \mathbb{R}^d$ is a function of $\nu(s')$.*

Appendix. C contains the full proof and derivation of $\phi$ and $\mu$ in terms of $\psi$, $\nu$, and $Z$.

To learn $\psi$ and $\nu$, we employ noise-contrastive estimation (NCE) (Ma & Collins, 2018; Gutmann & Hyvärinen, 2010; 2012), which enables optimization of unnormalized statistical models without explicitly computing the partition function. In this way, we sidestep the intractable computation of

$Z(s, a)$ in Eq. 6 by solving

$$\min_{\psi, \nu} \; \mathbb{E}_{\substack{(s,a) \sim \tau^e, s' \sim P(\cdot|s,a), \\ s_i'' \sim \rho}} \left[ \log \frac{\exp(\psi(s,a)^\top \nu(s'))}{\exp(\psi(s,a)^\top \nu(s')) + \Sigma_{i=1}^k \exp(\psi(s,a)^\top \nu(s_i''))} \right], \quad (7)$$

where $s'$ denotes a positive sample drawn from the transition distribution $P(\cdot|s,a)$, and $s_i''$ for $i = 1, \ldots, k$ are negative samples from marginalized state distribution $\rho(s) = \int \tau^e(s,a)da$.

*Connection to behavioral dynamics.* With simple algebra, we obtain the quadratic potential function $P(s'|s,a) \propto q(s') \exp\left(\|\psi(s,a)\|^2/2\right) \exp\left(-\|\psi(s,a) - \nu(s')\|^2/2\right) \exp\left(\|\nu(s')\|^2/2\right)$ from Eq. 6. By enforcing unit-norm constraints $\|\psi(s,a)\|^2 = \|\nu(s')\|^2 = 1$, assuming $Z(s,a)$ as a constant and $q(s')$ as uniform distribution, as well as taking $\nu$ to be the identity map, we obtain a generalized Gaussian form: $P(s'|s,a) = \frac{1}{Z} \exp\left(-\|s' - \psi(s,a)\|^2\right)$, which aligns with commonly adopted assumptions in animal behavior modeling discussed earlier. Thus, Eq. 6 offers a more general framework that extends traditional dynamics models for studying behavior. Moreover, in practice, using Eq. 6 results in a unimodal distribution over $s'$, which closely matches the empirical structure observed in $P(s'|s,a)$ from animal behavior data. Thus, while Eq. 2 is theoretically valid in continuous domains, we adopt the parameterization in Eq. 6 for continuous state and action spaces, as it more effectively supports the learning of motif representations underlying animal behavior.

**Motif-based policy learning.** After we obtain the representation $\psi(s,a)$ and $\nu(s')$ from Eq. 7, we could theoretically get motif sets $\phi(s,a)$ expressed as the function of $\psi(s,a)$ and a normalizing term $Z(s,a)$ (see Appendix. C). However, since in practice $Z(s,a)$ remains intractable, even with the optimal $\psi$ it is still hard to obtain $\phi$ exactly. Therefore, we introduce a mapping $f : \psi \to \phi$, parameterized with a neural network, and learn it via policy learning. Our aim is to learn a function $f$ so that $\phi = f(\psi)$ yields optimal basis functions of policy that best account for the animal behavior data. By applying $\phi = f(\psi)$ to Eq. 3, we obtain $Q(s,a) = f\left(\psi(s,a)\right)^\top u$.

As mentioned earlier, learning both $f$ and $u$ using the MLE objective in Eq. 5 becomes intractable for continuous data, as the denominator involves integration over an unbounded continuous action space. This brings us back to the challenge of estimating an unnormalized energy function, $\pi(a|s) \propto \exp(Q(s,a)) \propto \exp(f(\psi(s,a))^\top u)$. Thus, it's reasonable to apply NCE here again,

$$\min_{f, u} \; \mathbb{E}_{\substack{(s,a) \sim \tau^e, \\ (s_i', a_i') \sim \tau^e}} \left[ \log \frac{\exp(f(\psi(s,a))^\top u)}{\exp(f(\psi(s,a))^\top u) + \Sigma_{i=1}^k \exp(f(\psi(s,a_i'))^\top u)} \right], \quad (8)$$

where $\{s, a\}$ are positive samples and $\{s_i', a_i'\}$ are negative samples.

### 3.3 UNDERSTANDING ANIMAL BEHAVIOR VIA MOTIF AND POLICY LEARNING

In this section, we discuss how motor motifs $\phi$ and motif weights $u$ can be used to describe animal behavior trajectories, particularly allowing $u$ to vary across tasks $t$ or time points $t$. As the motif coefficients, $u(t)$ would dynamically modulate the influence of each motif on the final policy. We consider two behavioral modeling scenarios: (1) discrete state-action spaces in a multi-task setting, and (2) continuous state-action spaces in a time-varying setting. The concrete results will be later shown in Sec. 4.2 and Sec. 4.3 respectively. In either case, $u(t)$ would be a matrix where each column corresponds to one weight for one task or time point.

*Scenario (1)*: Consider a mouse navigating a maze (Rosenberg et al., 2021), where trajectories are discretized into a finite state space (locations) and actions are discrete (up, down, left, right, stay). We assume the animal switches between $T$ strategies, each associated with a unique reward, that guide its navigation, with the timing of each reward condition known from Ke et al. (2025). We first learn shared motifs $\phi(s,a)$ using Eq. 2, then fit task-specific policies $\pi(a|s,t)$ using Eq. 5, where each task $t$ corresponds to one of $T$ strategies. This yields $T$ sets of weights $u(t)$, one per task, while sharing a common motif set across tasks.

*Scenario (2)*: A representative case is a freely behaving mouse (Wiltschko et al., 2015; Weinreb et al., 2024), where the state is defined by pose keypoints and the action by state change–both continuous. We first learn $\psi(s,a)$ from pose trajectories using Eq. 7. Assuming the policy evolves smoothly over time, we learn $u(t)$ and $f$ via Eq. 8, yielding the motor motif $\phi(s,a) = f(\psi(s,a))$ and time-varying policy $\pi(a|s,t)$. Unlike models with abrupt discrete switches, this continuous-time formulation captures gradual behavioral changes more faithfully over long pose sequences.

Beyond capturing smoothly time-varying motif compositions, our framework allows multiple motifs to be active simultaneously. Since the policy is defined as $\pi(a|s,t) \propto \exp\big(\phi(s,a)^\top u(t)\big)$, each action is a generalized linear combination of basis motifs weighted by $u(t)$, enabling overlapping and composable behaviors. For instance, back grooming may blend grooming and turning back, while side grooming mixes grooming with turning side—recruiting different motifs concurrently. Unlike discrete switching-state models, which assign one behavior per state, our continuous motif-based approach provides a more flexible and interpretable representation of complex pose dynamics and, to our knowledge, is the first to offer a fully compositional and continuously time-varying description of animal trajectories. In Sec. 4, we would show (1) what motifs we have learned, and (2) how they are used to construct the final policy, on one simulation datasets and two real animal behavior datasets.

### 3.4 REWARD RECOVERY

After estimating $u(t)$ as the motif weights for policy construction, we can further infer $w(t)$ for reward representation $r(s,a,t) = \phi(s,a)^\top w(t)$ as $w(t) = u(t) - \gamma \int V(s',t)\,\mu(s')\,q(s')\,\mathrm{d}s'$, where $V(s,t) = \log \sum_a \exp Q(s,a,t)$. This allows us to recover the time-varying reward function $r(s,a,t)$ used by animals. Recovering the internal reward function aligns with the goals of inverse reinforcement learning (IRL) (Ziebart et al., 2008), where both the policy and underlying reward are inferred from demonstrations. In the context of animal behavior (Ke et al., 2025; Zhu et al., 2024; Ashwood et al., 2022), identifying such rewards offers insight into the internal motivations driving behavior and provides a window into animal cognition and decision-making processes. Since $V(s,t)$ can only be easily computed in closed form in discrete settings, we validate our method by visualizing the inferred rewards in the first two experiments, where the state-action space is discrete and finite.

## 4 EXPERIMENTS

### 4.1 APPLICATION TO SIMULATED DATA IN A MULTI-TASK GRIDWORLD

The gridworld consists of a $3 \times 3$ lattice with nine discrete states, and each state allows four possible actions: Up, Down, Left, and Right (Fig. 2A). In task $i$, a high reward is assigned to the $(s,a)$ pairs that move toward the location $i$. Fig. 2C (left) shows the ground truth of reward functions for all nine tasks. In each episode, the agent starts from a random start state and must navigate to the task-specific location $i$. See Appendix. D for more details for data generation.

As described in Sec. 3.1, we learn a set of latent motifs and use them to construct the task-specific policy $\pi(a|s,t)$ for each task $t \in \{1,...,9\}$. Because the computational complexity only scales linearly with the number of motifs, to cover the motif space as much as possible, we select a large number for motif dimension $d = 64$. (See Appendix. J.1 for discussions of motif dimensions.) Visualizations of these motifs are shown in the Appendix. D. Using the learned motifs, we recover the policy and further infer the reward function, as described in Sec. 3.4, with the form: $r(s,a,t) = \phi(s,a)^\top w(t)$. This recovered reward (Fig. 2C, right) closely matches the ground truth, achieving a Pearson correlation coefficient of 0.96, indicating that the learned motifs are sufficient for accurately reconstructing the reward function from behavior data.

To better interpret the learned motifs and their role in reward composition, we apply principal component analysis (PCA) to the $\phi$ matrix and find that only 8 principal components capture most of the variance (Fig. 2B). Therefore, we visualize the top 8 PC motifs and their corresponding task-specific coefficients $w(t)$ in Fig. 2D, treating them as basis vectors spanning the motif space. The PC motifs exhibit interpretable patterns. For instance, in motif 0, $(s,a)$ pairs leading to the bottom-left grid have strong positive values, while those leading to the middle-right grid have strong negative values. This motif corresponds to moving away from the middle-right grid toward the bottom-left. Examining the $w$ matrix, we see that motif 1 has a strong positive weight for task 8, consistent with the goal of moving toward the bottom-right corner in that task. In contrast, motif 0 contributes negatively to task 8, as it promotes movement toward the bottom-left and away from the goal. Similar interpretations can be made for other motifs and tasks.

In this gridworld experiment, we successfully recover reward functions from behavior trajectories, and, importantly, the learned $\phi$ and $w$ are effectively deployed in different task settings.

### 4.2 APPLICATION TO ANIMAL NAVIGATION BEHAVIOR

**Dataset and model setup.** We next evaluate our method on a real animal behavior dataset from Rosenberg et al. (2021). In this experiment, a thirsty mouse is trained to navigate in a binary-tree maze (Fig. 3A), starting each trial from the central home cage and attempting to reach a water port at one leaf node. The state space is defined by the mouse's location on the tree. The actions include moving

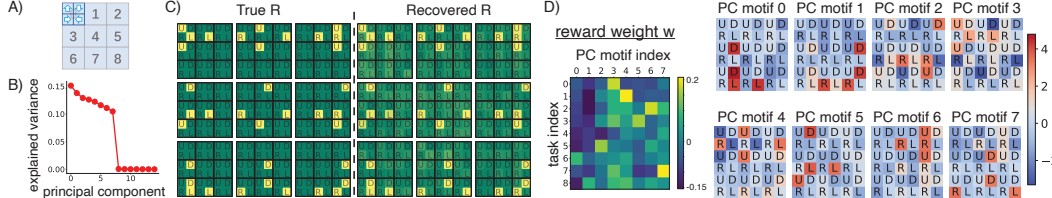

Figure 2: **A.** State-action map: each of the nine grids is divided into four cells representing action values (Up, Down, Left, Right). In task $i$, high reward is assigned to $(s, a)$ pairs moving toward the $i$th location. **B.** Explained variance of the top 15 principal components; variance drops near zero after PC7. **C.** Left: true rewards for all 9 tasks (yellow = high, green = low). Right: recovered rewards. **D.** Reward weight $w$ and top 8 PC motifs from the $\phi$ matrix. Reward weights indicate the contribution of the top 8 PC motifs to each task. In the PC motif plot, red indicates positive feature values, blue indicates negative.

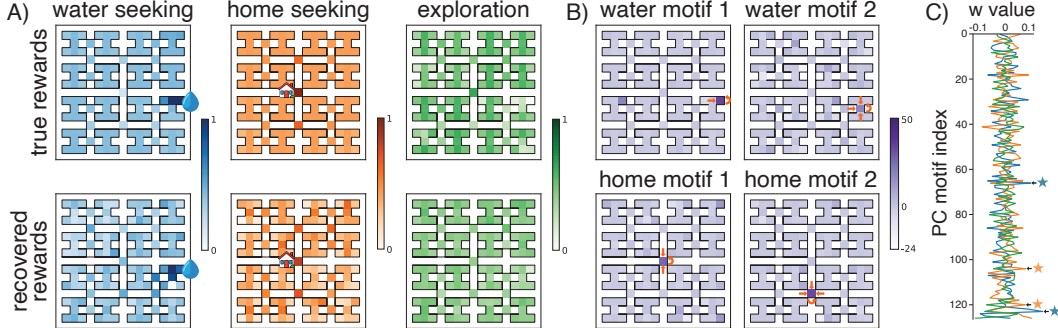

Figure 3: **A.** True and recovered rewards for the three tasks. **B.** Two dominant motifs for the water and home tasks respectively. Each motif indicates that taking a specific action (orange arrow) toward the dark purple state yields a high value. **C.** $w$ values for all tasks, colored by task; blue stars highlight high-weight motifs for water seeking, and orange for home seeking, all shown in (B).

to its left parent, right parent, left child, and right child. Although the mouse's behavior is primarily driven by water foraging, it also exhibits exploration of unvisited areas and returns to the home cage for shelter. This complex behavior cannot be captured by a single reward function. Studying this dataset allows us to discover motifs shared across multiple reward functions and policies. This in turn tests whether complex navigation behavior, under multiple competing motivations, can be distilled into a small set of reusable motifs that provide interpretable insight into animal decision-making.

We first apply the segmentation algorithm from Ke et al. (2025) to divide long behavior trajectories into three interpretable tasks: water seeking, home seeking, and exploration. The algorithm could also infer reward maps for these tasks, which are shown in Fig. 3A (top row). While the mouse's true internal reward functions remain unknown, we treat these inferred rewards as effective ground truth since they generate policies that closely replicate the observed behavior. Thus, we are able to segment long navigation trajectories into task-specific episodes, yielding a multi-task dataset with discrete $(s, a)$ pairs, analogous to our simulated gridworld setting. To uncover a task-agnostic set of motifs capable of constructing all three reward functions, we apply our model as described in Sec. 3.1. Again, to cover the motif space as much as possible, we set the number of motifs $d = 127$. (See Appendix J.1 for more details on the selection of $d$)

**Results.** To assess model performance, we estimate the task-specific weights $w(t)$ and reconstruct the reward function as $r(s, a, t) = \phi(s, a)^\top w(t)$. The recovered reward functions align closely with the ground truth (Fig. 3A, bottom row). In the water-seeking task, the recovered reward has peaks near the water port and along the path leading to it. During home-seeking, a distinct peak appears at the home cage. When exploring, the reward is nearly uniform across the maze, with a notable dip at the water port, suggesting the mouse temporarily suppresses water motivation to explore other areas.

We further visualize the learned motifs by applying PCA to obtain PC motifs. See the Appendix. E for raw features of motifs. Fig. 3B displays two top-contributing motifs for the water and home tasks respectively, selected based on the peak of linear weights $w$ (Fig. 3C). The water-related motifs promote movement toward the water port, while the home-related motifs guide navigation to the cage. Notably, the motifs important for one task have minimal or negative contributions to the other

(Fig. 3C), indicating clear functional specialization. Fig. 3C also shows non-task-specific motifs with similar weights in both water- and home-seeking tasks.

These results show that our model not only recovers multiple reward functions from real behavior but also learns interpretable motifs whose contributions to each task are distinct and behaviorally meaningful. Unlike previous reward discovery on this dataset (Ashwood et al., 2022; Ke et al., 2025), which cannot identify such motifs, our approach reveals how reward maps can be decomposed into smaller, reusable action components. This decomposition offers a mechanistic view of how local decision processes combine to produce strategies such as water-seeking or home-seeking.

### 4.3 APPLICATION TO ANIMAL FREE-MOVING BEHAVIOR

**Dataset and model setup.**   To verify the generalizability of our method to continuous scenarios, we apply our method to a continuous dataset of free-moving mouse behaviors (Weinreb et al., 2024) to extract motor motifs and analyze fine-grained pose dynamics. This dataset contains the keypoint coordinates of eight mouse body parts, including the head, the nose, both ears, and four spine nodes. Each dimension of the state corresponds to either $x$ or $y$ coordinate of a body part. The action at each timestep is defined as the velocity of state, $a_t = (s_{t+1} - s_t)/\delta t$. We set the time interval to $\delta t = 1$ ($1/30s$ in the original dataset). We formulate the data using a continuous MDP. Studying this dataset allows us to ask whether free-moving behaviors, which often appear as mixtures of grooming, locomotion, and postural adjustments, can be represented as combinations of a compact set of task-agnostic motor motifs.

As outlined in Sec. 3.2, we use NCE to learn the motif representation $\phi(s, a)$ and the time-dependent weights $u(t)$. To ensure temporal smoothness in the learned weights $u(t)$, we place a Gaussian random walk prior over the trajectories: $u(t) \sim \mathcal{N}(u(t-1), \sigma^2 I)$. To perform more stable optimization, we optimize $\phi(s, a)$ and $u(t)$ using coordinate descent, updating them alternately. For this dataset, our focus is on understanding the learned policy structure, which reflects the dynamics of animal poses. Therefore, we do not perform IRL in this setting and instead concentrate on interpreting the learned motif representation $\phi$ and the temporal weights $u(t)$. With respect to the choice of the number of motifs, we find in practice that the performance grows more slowly once past $d = 64$, so we choose this as an optimal number. See Appendix. F for more training details. See Appendix. J.1 for details on the selection of $d$.

We compare our MCD method with two representative behavior segmentation approaches: (1) **Keypoint-MoSeq** (Weinreb et al., 2024), as a representative for switching-dynamics-based segmentation methods; and (2) **SemiSeg** (Whiteway et al., 2021), as a representative for clustering-based behavior segmentation methods. Note that Keypoint-MoSeq is regarded as a SOTA approach, because it extends the autoregressive hidden Markov model (AR-HMM) and MoSeq (Wiltschko et al., 2015), and has been shown to outperform B-SOiD (Hsu & Yttri, 2021), VAME (Luxem et al., 2022), and MotionMapper (Berman et al., 2014). We also include **OPAL** (Ajay et al., 2021), a representative autoencoder-based motif learning algorithm from robotics, as a baseline to highlight the advantages of our method and why robotics approaches are ill-suited for behavioral segmentation in neuroscience.

**Results.**   We evaluate performance using the area under the Receiver Operating Characteristic (ROC) curve (AUC), which quantifies the model's ability to distinguish positive from negative samples. AUC is chosen because it allows direct comparison between our unnormalized energy function and Keypoint-MoSeq's likelihood score. Given $(s, a), (s', a') \sim \tau^e$, we define positive samples as $(s, a)$ and negative samples as mismatch pairs $(s, a')$. With respect to the choice of the prediction score, for Keypoint-Moseq, we use the action log-likelihood. For MCD, we use the negative energy function $\phi(s, a)^\top u(t)$. For SemiSeg, we assume action variance=1 and use the Gaussian log-likelihood of actions. For OPAL, we use the action log-likelihood.

We repeat the experiment 10 times and report the results as box plots in Fig. 4A. Our model achieves the highest AUC on both training and test sets (paired t-test, $p < 0.05$ for every baseline), demonstrating the strongest ability to distinguish positive from negative samples. This suggests that MCD accurately captures time-varying pose dynamics through smoothly evolving motifs, while other models fail. Keypoint-MoSeq's reliances on discrete switching syllables produce a coarser representation of the underlying complexity, and the autoencoders in the other two models have weaker expressive ability.

Beyond quantitative comparisons, we also qualitatively visualize and interpret the key motifs $\phi(s, a)$ associated with example pose dynamics. For MCD, we examine the time-varying weights $u(t)$ of a

long animal behavior video (length=250) and choose five example animal behavior clips (length=5) to check the interpretability of the result (Fig. 4B). For our model, each clip is characterized by a unique combination of motor motifs. For each clip, we show the top 1-2 most dominant motifs. For each motif, we display the animal's skeleton with red arrows showing the motion field, computed by averaging the actions that most strongly activate that motif. The movement semantics of each motif are labeled above the visualizations. A more comprehensive visualization of all learned motifs is included in the Appendix. F. For comparison, we run other models on the same behavior video, showing the latent representations by SemiSeg (Fig. 4C) and OPAL (Fig. 4D). To show clearer results for SemiSeg and OPAL, we further run KMeans (k=10) on the first 10 PCs of the latents throughout the video and show the segmentation result at the bottom of the latent representations (Fig. 4C, D). We also show the behavioral syllable segmentation produced by Keypoint-MoSeq in Fig. 4E.

By combining the discovered motifs with real animal behavior, we assess the interpretability of each motif in the five clips (Fig. 4B). First, the right-turn behavior in clip 1 is captured by the dominance of two rightward motifs (motif 1 and motif 2). A stronger movement at the head is reflected by a higher value of motif 1. Clips 2, 3, and 4 show the mouse turning right, pausing to groom its head and ears, and then continuing to turn right. The alternate dominance of motif 3 and motif 4 aligns well with the behavior dynamics. Clip 5 shows a simultaneous behavioral mixture of moving forward and sniffing, and is captured by the equal strength of motif 5 and motif 6. Across all motifs, motif 4 appears across clips 2, 3, and 4, showing its general utility. The transitions and mixtures of behaviors are effectively reflected in the learned motifs and their temporal weights $u(t)$.

However, when we examine the segmentation and latent produced by other methods (Fig. 4C-F), we find inconsistencies. In Fig. 4C, alternate dominant behavior patterns in Clip 2-4 in the video are not reflected in its motif weights during this period, as the only dominant motif is the yellow-green one. In Fig. 4D, the motif weights are too dense to interpret. The segmentation results at the bottom of Fig. 4 C and D do not even have repeated behavior patterns and thus could not be seen as a reasonable behavior motif representation. For Keypoint-Moseq, in Fig. 4E, F, clear rightward turning in clip 1 is barely visible in syllable 3 to which it is assigned. Clips 2 and 3 are both assigned to syllable 2, even though clip 2 shows pure turning right while clip 3 is dominated by grooming movements. For clip 5, the mixture of fast moving forward and sniffing is not reflected in syllable 2.

Taken together, these results show that compared with similar approaches, MCD provides a more accurate interpretation of pose dynamics and could capture more complex behaviors through a compact, task-agnostic set of motor motifs. This offers us a detailed perspective on how intricate behaviors emerge from the dynamic combination of fundamental motor motifs.

## 4.4 EVALUATION ON A HUMAN-ANNOTATED DATASET.

Last, we verify our method on a supervised behavior label dataset. Due to the substantial cost and labor demands of manual annotation, large-scale, high-quality animal behavior datasets with human labels remain limited. Commercial annotation tools (e.g., HomeCageScan) are also expensive, further constraining availability. To support quantitative evaluation, we therefore included a human-annotated subset from the freely moving animal behavior dataset we mentioned last section. Specifically, we selected 200 short video clips (10 recording sessions; in each session we randomly sampled 20 clips, with 200 frames each). Each clip was manually annotated with behavior segments for each frame, resulting in 200 per-frame labels across six categories: `Walking`, `Sniffing/Grooming`, `Turning left`, `Turning right`, `Rearing`, and `Resting`. Manual annotation required approximately 40 hours of expert effort.

As before, we evaluated four models on this dataset: MCD, Keypoint-MoSeq, SemiSeg, and OPAL. Because the latent variables or syllables inferred by these models do not necessarily align with human-defined labels, we trained an additional decoder to map model-specific latent representations to the ground-truth categories. For each model, the decoder input was the motif weights $u(t)$ (MCD), the inferred one-hot syllable (Keypoint-MoSeq), or the latent embedding $z$ (SemiSeg and OPAL). The decoder was implemented as a two-layer neural network with ReLU activation, optimized using the Adam optimizer and cross-entropy loss (learning rate 0.001), following standard settings in the scientific computation package `scikit-learn`. The resulting classification accuracies on the held-out test set (Fig. 4G) demonstrate that MCD substantially outperforms all baselines, achieving

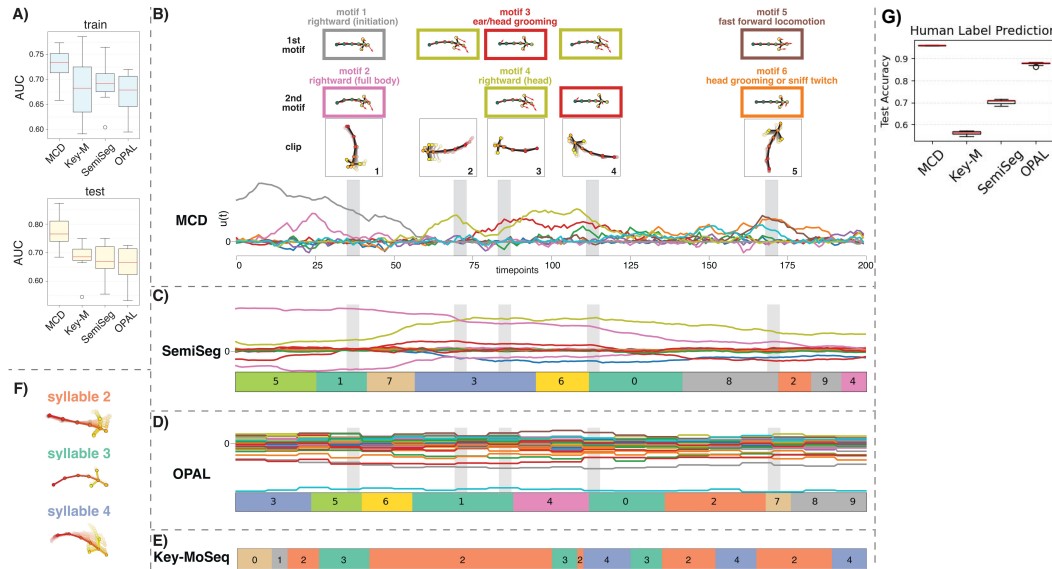

Figure 4: **A.** AUC on training and test sets. We take an example behavior video and run our algorithm. We visualize the motif weights $u(t)$ and show the representative motifs in **B**. For the baseline SemiSeg, we show the latent skills and segmentation results in **C**. For the baseline OPAL, we show the latent skills and segmentation results in **D**. Then we show the segmentation results of Keypoint-MoSeq (Weinreb et al., 2024) in **E** and the representative motifs/syllables in **F**.

near-perfect accuracy (0.96). This indicates that the motifs recovered by MCD align better with actual animal behavior dynamics.

## 5 DISCUSSION

Several limitations remain to be addressed in future work. First, the accuracy of inferred motifs is sensitive to input data quality, as occlusions or tracking errors can degrade performance. Additionally, while the framework uncovers abstract motor primitives, establishing direct correspondences between these learned "motifs" and specific neural dynamics still requires further experimental validation.

## ETHICS STATEMENT

Beyond advancing animal behavior research, MCD has broader implications. Positively, a better understanding of motor control mechanisms could, for instance, inform new treatments for movement disorders or inspire more adaptable AI. On the other side, extending these principles to model human behavior carries ethical risks, such as perpetuating or amplifying societal biases present in training data. A robust ethical framework is essential to mitigate such risks in the development and application of these technologies.

## REPRODUCIBILITY STATEMENT

We have taken several steps to ensure the reproducibility of our results. All source code for model training and evaluation is included in the supplementary material, allowing independent verification and replication of our experiments. The complete set of hyperparameter values is documented in the appendix. Additionally, the data preprocessing procedures and evaluation protocols are described in the main text. These resources provide sufficient information for reproducing the results reported in this paper.

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

## A  LLM USAGE

In preparing this manuscript, we employed a large language model (OpenAI ChatGPT, GPT-5) as a writing assistant. The model was used exclusively for polishing English grammar, improving clarity, and suggesting more natural phrasing in certain sections of the text. All scientific content, experimental design, analyses, and interpretations were conceived, written, and verified by the authors. The LLM was not used to generate original research ideas, analyses, or results. To ensure accuracy, all model-suggested edits were carefully reviewed and, where necessary, modified by the authors.

## B  HYPERPARAMETER SETTING

We train MCD using the following hyperparameters:

**General hyperparameters.**

- Discount factor: $\gamma = 0.99$
- Number of epochs: $1 \times 10^6$
- Batch size: 256

**Motif representations.**  The motif representation $\phi(s,a) \in \mathbb{R}^d$ and $\mu(s') \in \mathbb{R}^d$ were adopted with different motif dimensions $d$ depending on the task:

- Gridworld: $d = 64$
- Animal navigation: $d = 128$
- Animal free-moving: $d = 64$

**Model architectures.**

- **Discrete version:** $\phi$ and $\mu$ are parameterized by one-hidden-layer neural networks (hidden size = 512).
- **Continuous version:**

    - $f$: no hidden layer
    - $\nu$: one hidden layer (hidden size = 512), with one normalization layer to ensure the L2-Norm of the output is 1.
    - $\psi$: no hidden layer, with one normalization layer to ensure the L2-Norm of the output is 1.
- In both cases, $u$ and $w$ are just two matrices, with each column corresponding to $u(t)$ or $w(t)$ at a specific task or timepoint.
- Activation function: For all networks, if it has at least one hidden layer, then all of its activation functions are Exponential Linear Unit (ELU).

**Learning rates.**

- **Discrete version:**
$$\phi : 1 \times 10^{-3}, \quad \mu : 1 \times 10^{-3}, \quad u : 3 \times 10^{-4}, \quad w : 3 \times 10^{-4}.$$
- **Continuous version:**
$$\psi : 5 \times 10^{-4}, \quad \nu : 5 \times 10^{-4}, \quad f : 3 \times 10^{-4}, \quad u : 3 \times 10^{-4}.$$
During testing, $u$ and $f$ are further optimized on the new sequence using gradient descent with learning rate $1 \times 10^{-3}$.

We train SemiSeg and OPAL using the following hyperparameters:

- Discount factor: $\gamma = 0.99$
- Number of epochs: $1 \times 10^6$
- Batch size: $4 \times 250$ (4 sequences, each of length=250 because this is an RNN-based inference model)
- Latent dimension: $d = 64$
- Learning rate: $1 \times 10^{-4}$ (tuned to get better results)

Besides, the following loss coefficients are shared across three models for interpretability results.

- Temporal smoothness Gaussian random walk loss: 10
- Sparsity L1-loss: 0.1

## C  APPROXIMATING ENERGY-BASED FORMULATION WITH LOW-RANK SPECTRAL DECOMPOSITION

In this part, we show in detail the connection between the EBM formulation (Eq. 6) and the low-rank spectral decomposition formulation (Eq. 2) of the transition kernel $P(s'|s, a)$.

From Eq. 6, by simple algebra, we obtain the quadratic potential function,

$$P(s'|s, a) \propto q(s') \exp\left(\|\psi(s, a)\|^2/2\right) \exp\left(-\|\psi(s, a) - \nu(s')\|^2/2\right) \exp\left(\|\nu(s')\|^2/2\right). \quad (9)$$

The term $\exp\left(-\frac{\|\psi(s,a) - \nu(s')\|^2}{2}\right)$ is the Gaussian kernel, for which we apply the random Fourier feature (Dai et al., 2014; Rahimi & Recht, 2007) and obtain the spectral decomposition of Eq. 6 as

$$P(s'|s, a) = \langle \phi_\omega(s, a), \mu_\omega(s')q(s') \rangle_{\mathcal{N}(\omega)}, \quad (10)$$

where $\omega \sim \mathcal{N}(0, I)$ is the frequency in the Fourier domain, and

$$\phi_\omega(s, a) = \exp\left(-i\omega^\top \psi(s, a)\right) \exp\left(\|\psi(s, a)\|^2/2 - \log Z(s, a)\right), \quad (11)$$

$$\mu_\omega(s') = \exp\left(-i\omega^\top \nu(s')\right) \exp\left(\|\nu(s')\|^2/2\right). \quad (12)$$

Note that Eq. 10 needs infinite $\omega$ to calculate the expectation. To connect it to finite dimension $\phi(s, a) \in \mathbb{R}^d, \mu(s') \in \mathbb{R}^d$, we use the Monte-Carlo method to approximate it with finite samples,

$$P(s'|s, a) \approx \frac{1}{M} \sum_{i=1}^{M} \phi_{\omega_i}(s, a)\mu_{\omega_i}(s')q(s'). \quad (13)$$

Introduce vectors $\phi(s, a)$ and $\mu(s')$ such that

$$\phi(s, a) := \frac{1}{\sqrt{M}} \left[\phi_{\omega_1}(s, a), \phi_{\omega_2}(s, a), ..., \phi_{\omega_M}(s, a)\right], \quad (14)$$

$$\mu(s') := \frac{1}{\sqrt{M}} \left[\mu_{\omega_1}(s'), \mu_{\omega_2}(s'), ..., \mu_{\omega_M}(s')\right]. \quad (15)$$

Then it's straightforward to see that,

$$\phi(s, a)^\top \mu(s')q(s') = \frac{1}{M} \sum_{i=1}^{M} (\phi_{\omega_i}(s, a)^\top \mu_{\omega_i}(s')) \approx P(s'|s, a). \quad (16)$$

Hence, Eq. 6 can, in principle, yield the motif representation introduced earlier.

# D  MULTI-TASK GRIDWORLD DATASET

## D.1  DATASET

To generate the dataset, we follow this procedure: 1) Use soft value iteration to compute the ground truth Q-function for each task: $Q(s, a, t) = r(s, a, t) + \log \sum_a \exp V(s, t)$; 2) Use the resulting Q-function to define the policy: $\pi(a|s, t) = \frac{\exp(Q(s,a,t))}{\sum_{a'} \exp(Q(s,a',t))}$ and sample trajectories accordingly.

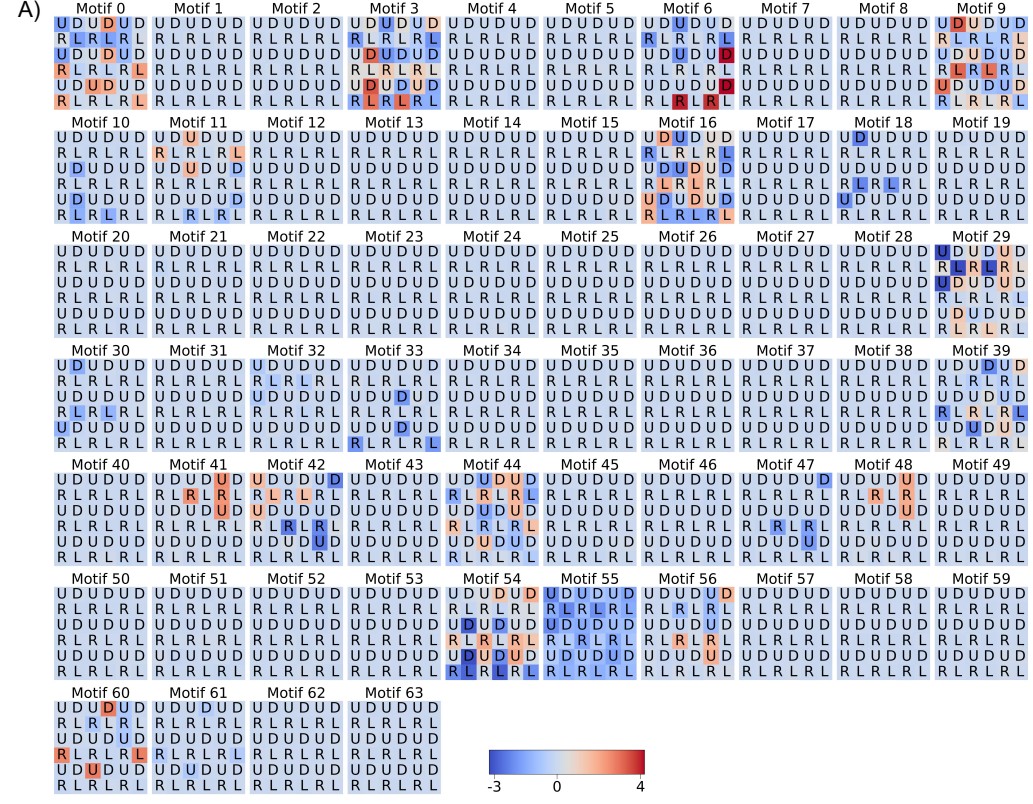

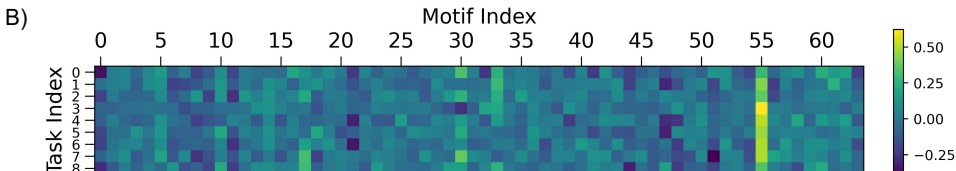

Figure 5: **A.** State-action maps for all 64 motifs. **B.** Reward weight $w$ for all 64 motifs.

## D.2  LEARNED MOTIFS

We visualize all original 64 motifs (Fig. 5) introduced in Sec. 4.1. It shows some meaningful patterns as mentioned before. For example, motif 0 assigns high values to those $(s, a)$ pairs leading to the middle-middle grid and the bottom-middle grid, and assigns low values to the up-left grid and the up-right grid. Thus, it is employed negatively in Task 0 (up-left reward) and Task 2 (up-right reward). However, Task 4 (middle-middle) didn't use this motif and used motif 39 negatively instead. The complex many-to-many relationship between motifs and tasks informs us of the redundancy in the original motifs, which inspires us to use PCA to analyze the principal components of the motif space and simplify the motif weights. It could be seen from the comparison between Fig. 2D and Fig. 5A that principal components are a less redundant description of the motif space.

# E ANIMAL NAVIGATION BEHAVIOR DATASET

## E.1 LEARNED MOTIFS

In the original motifs of the labyrinth environment, multiple $(s, a)$ pairs are simultaneously activated, so it is rather hard to analyze which $(s, a)$ pairs are the most important ones that could represent the focus and function of the motif. Given the redundancy of the motif sets, as in Appendix. D, we perform PCA to analyze the principle components of the motif space and simplify the motif representation. To show the effect of PCA, we plot one motif (motif 0) before (Fig. 6A right) and after PCA (Fig. 6B right). Basically, we only want to show the most important pairs in one map and do not want low-value pairs to disturb the visualization. To determine how many $(s, a)$ pairs are important, we sort the $(s, a)$ pairs based on the value $\phi(s, a)$ in motif 0, i.e., the first feature of the output of $\phi(s, a)$ (Fig. 6A left and B left). It could be seen straightforwardly that after PCA, the motif becomes more concentrated on several $(s, a)$ pairs. We calculate the mean $\mu$ and variance $\sigma$ across all motifs and all dimensions and take $\mu + \sigma$ as the threshold, above which $(s, a)$ pairs are deemed the most important ones and are shown on the right. We show 80 pairs before PCA and 8 pairs after PCA. The number of the most important pairs in each motif is called the "effective dimension." The effective dimension is calculated across all motifs (Fig. 6C). Paired t-test ($p = 1.3 \times 10^{-51}$) shows that there exists a significant decrease of effective dimensions after PCA. So the map becomes more distinct and functionally separated.

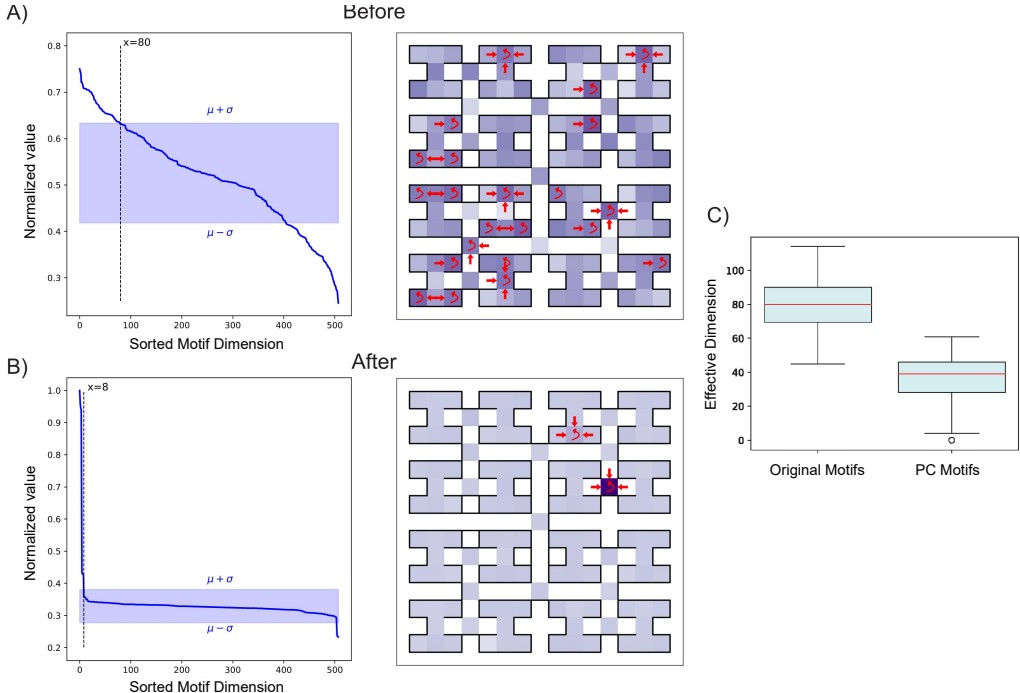

Figure 6: **A.** Left. The value for each $(s, a)$ pair in motif 0 before PCA. Right. The most important $(s, a)$ pairs. **B.** Left. The value for each $(s, a)$ pair in motif 0 after PCA. Right. The most important $(s, a)$ pairs. **C.** Boxplot for the effective dimensions before and after PCA.

# F ANIMAL FREE-MOVING BEHAVIOR DATASET

## F.1 DATASET AND TRAINING

We split the full dataset into training and test trajectories in an 8:2 ratio. We first learn both $f$ and $u(t)$ on the training set. Then, given the learned $f$, we estimate $u(t)$ on the test set. Here, $f$ is a time-invariant model parameter shared across all time, while $u(t)$ is a time-dependent variable that must be inferred separately for each test trajectory and cannot be transferred from training.

## F.2 LEARNED MOTIFS

We show all motifs learned from the 200-timestep video clip of the free-moving mouse mentioned in Sec. 4.3. We have completed the visualization of those motifs that were previously omitted due to their perceived lack of importance. Due to the increased number of displayed motifs, we had to renumber each motif. We show the present motif number above the motif motion field figure, and previous numbers (if applicable) in the parentheses following the present number.

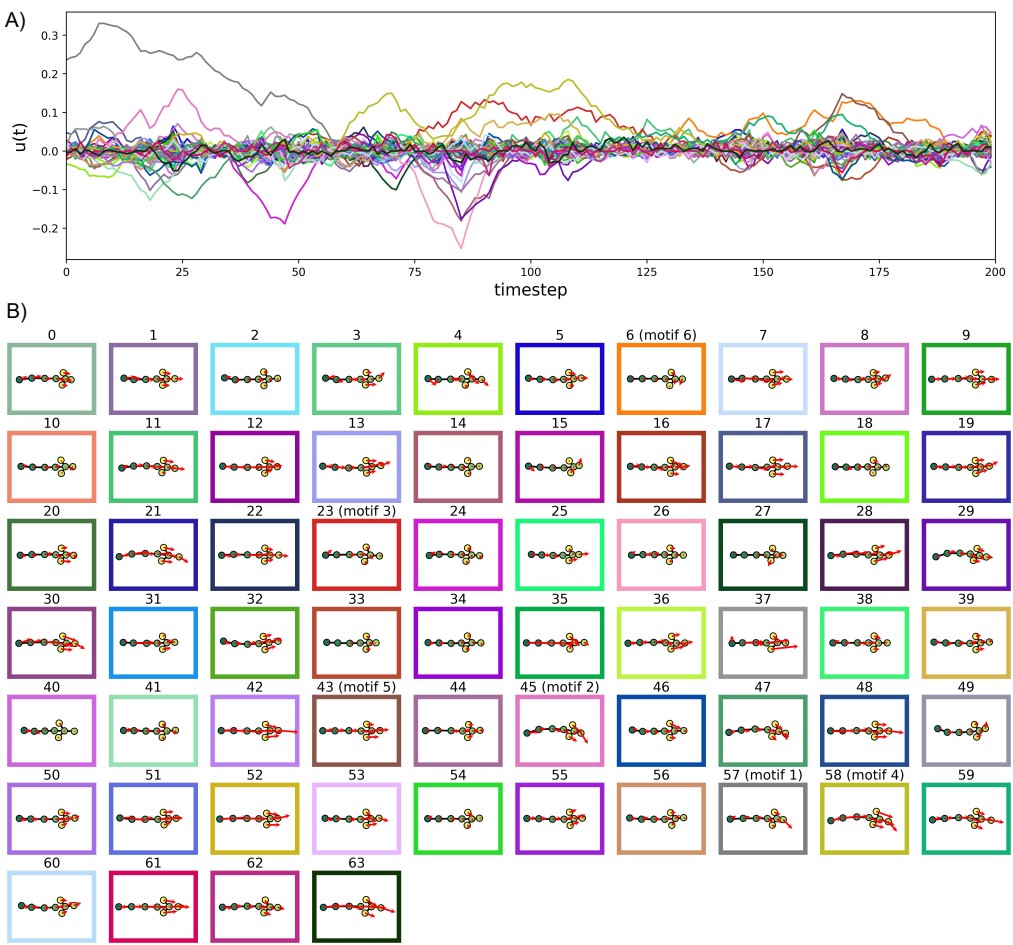

Figure 7: **A.** Policy weight $u(t)$ for all 64 motifs. **B.** The motion field for all 64 motifs learned from the video, computed by averaging the states and actions that most strongly activate each motif.

# G CONNECTIONS TO RELEVANT ALGORITHMS IN RL

**Motor primitives learning.** Animal behavior research has focused on identifying motor primitives directly from large-scale recordings. Supervised classification (Marks et al., 2022; Segalin et al., 2021), clustering-based analysis (Hsu & Yttri, 2021; Berman et al., 2014), and HMM-based methods (Wiltschko et al., 2015; Weinreb et al., 2024; Luxem et al., 2022; Whiteway et al., 2021) label behavior into discrete syllables and analyze neural correlates of transitions. While these approaches reveal stereotyped motifs such as walking or grooming, they typically treat primitives as rigid and exclusive, overlooking the continuous, compositional structure of behavior and its adaptation across contexts.

In robotics, movement primitives have long been studied (Paraschos et al., 2013; Saveriano et al., 2023; Lioutikov et al., 2017). ProMP (Paraschos et al., 2013) extracts primitives from demonstrations but requires labeled skills/motifs, while ProbS (Lioutikov et al., 2017) jointly infers behavioral segmentation and a primitive library via Expectation-Minimization (EM), enabling unsupervised skill discovery. These approaches resemble dynamics-based behavior segmentation in animal behavior research.

All methods mentioned here lack an RL perspective that links primitives to policies and reward-driven behavior. Our framework instead leverages offline RL-based imitation learning to uncover motif-based policies that capture both modular primitives and their sequential composition, and provides a principled way to explain why behaviors occur, not just how they unfold, and how reusable motifs/skills contribute to them in a generative decision-making process.

**Offline Imitation learning.** Offline imitation learning presents the problem of learning a policy from fixed demonstrations when access to environments is impossible. Simple behavior cloning can be performed offline, but fails to generalize well in some cases because it does not consider dynamics or environment structure limits. ValueDICE (Kostrikov et al., 2020) considers the dynamics of the training data and learns a policy that minimizes the KL-divergence between the state-action occupancies generated by the policy and of the original dataset. But the adversarial optimization of policy and Q-functions introduces instability in training. IQL (Garg et al., 2021) avoids adversarial training by directly parameterizing the policy in terms of the $Q$-function, $\pi(a|s) = \exp(Q(s, a))/Z$. However, all of these algorithms learn policies as unstructured functions. As the animal behavior structure is highly modular and stereotyped, it is more appropriate to employ a hierarchical motif-based policy to model the data.

**Motif/Skill discovery.** Unsupervised skill discovery (DIAYN (Eysenbach et al., 2019), BeCL (Yang et al., 2023), DADS (Sharma et al., 2020), InfoGAIL (Li et al., 2017), Directed-InfoGAIL (Sharma et al., 2019), SkillBlender (Kuang et al., 2025), ASE (Peng et al., 2022)) to find a high-level abstraction of actions has been an effective strategy for online RL and imitation learning. However, they are limited to online settings when the skills can only be refined through interacting with the environment, which restricts their application to large-scale offline datasets. Recent offline skill discovery methods include OPAL (Ajay et al., 2021), SPiRL (Pertsch et al., 2021a), and SkiLD (Pertsch et al., 2021b) which use an autoencoder to encode trajectories into latent skills $z$; and PARROT (Singh et al., 2021) which uses a flow-based model to learn a behavior prior. In these models, the latent $z$ is later used to generate the policy $\pi(a|s, z)$ in a non-linear way. Our paper, instead, employs a generalized-linear structure of the policy $\pi(a|s, z) \propto \exp(\phi(s, a)^\top u(z))$ which provides better interpretability than the policy network. This interpretability is essential for addressing downstream neuroscience questions.

**Linear structure of environment/policy.** Another line of work (HILP (Park et al., 2024), FB (Touati & Ollivier, 2021), USFA (Borsa et al., 2019), RaMP (Chen et al., 2023)) based on successor features (SF) also uses a generalized linear structure to model the motif-based policy as $\pi(a|s, z) \propto \exp(F(s, a, z)^\top z)$ where $F(s, a, z)$ is SF under a certain motif $z$. The main concern is that they cannot separate motif representation $z$ from state-action representation in SF, while our work can (Eq. 4). Therefore, their motifs depend on the specific task or timepoint, while our motifs/skills are general representations shared across tasks and timepoints. The idea that task-agnostic motifs are combined adaptively to form new policies aligns better with the need of interpretability and scientific discovery: we would like to look for neural signals responsible for relatively fixed time-agnostic behavior patterns, to help us better understand the animal behavior.

TRAIL (Yang et al., 2022) adopts a linear decomposable environment as MCD when learning the latent skills. However, the latent-conditioned policy $\pi(a|s, z)$ is not linear, but parametrized by a neural network. SkillBlender (Kuang et al., 2025) adopts a linear decomposable policy, and uses linear combinations of lower-level controller outputs as the final actions $a_t = \sum_i k_i a_t^i$, while we use the linear combinations of different motifs to generate the final state-value functions $Q_t(s, a) = \sum_i \phi_i(s, a)u_t^i$ rather than direct actions. We believe a naive mixture of the low-level controller is less biologically realistic in modeling animal behavior than mixtures of state-value functions. The latter can find supporting evidence in neuroscientific literature (Makino, 2023). Besides, their transition kernel $P(s'|s, a)$ is not linear.

Compared to them, our work assumes generalized linear structures for both the policy and the environment, providing better interpretability. This shared motif is more fundamental in revealing the basic structure of the animal's intentions.

**Our contributions.** Most existing skill-learning methods do not apply to the behavior understanding scenario considered here. We study offline data without supervision or task annotations—no explicit goals, labels, or trajectory segmentation—only long, unstructured behavior sequences. The challenge is threefold: (1) to discover the basic motor skills, (2) to determine how these skills compose spatially and temporally within long trajectories in an offline setting, and (3) to ensure interpretability for scientific discovery. These are nontrivial problems beyond the reach of existing skill- or motor-primitive approaches. To our knowledge, this is the first work to introduce offline RL-based imitation learning for behavior segmentation, yielding interpretable skill representations and their compositions.

# H  IDENTIFIABILITY DISCUSSION

In this section we clarify the notion of identifiability and how it applies to our motif-based continuous dynamics (MCD) model. A parameterization is said to be *identifiable* if different parameter values lead to different model predictions. Conversely, if multiple distinct parameterizations yield identical likelihoods or value functions, the model is only identifiable up to an equivalence class (e.g., scaling, rotation, or permutation).

**Our model is identifiable at the subspace level.**   In MCD, both the transition operator and the action-value function admit the spectral form

$$P(s'|s,a) = \phi(s,a)^\top \mu(s'), \qquad Q(s,a) = \phi(s,a)^\top u.$$

Although individual components of $\phi$, $\mu$, and $u$ may be rescaled by an invertible linear transformation without altering $P$ or $Q$, the *subspace spanned by the motif functions* is uniquely determined (Ren et al., 2023). This is analogous to PCA, where the principal subspace is identifiable even if the basis vectors within that subspace are not uniquely determined. In practice, they converge well.

Moreover, in the continuous version, the identifiability of the new representations $\psi, \nu$ up to rescaling has been guaranteed by the normalization operation before the output (Appendix B), which also ensures the trainability of the contrastive learning algorithm.

**Comparison to HMMs and latent variable models.**   Classical behavioral segmentation methods (e.g., Keypoint-Moseq as an HMM variant, SemiSeg as an autoencoder variant) are generally not identifiable without strong additional assumptions. Different transition matrices, emission matrices, or latent dynamics parameters can produce indistinguishable observation distributions. These models typically suffer from label-swapping and more severe transform equivalences in the latent space, meaning that even the latent subspace is not uniquely determined. In contrast, our method fixes the ambient representation through the observed state–action pairs and learns a function class whose span is uniquely tied to the transition operator and value function. Thus, while our motif basis is identifiable only up to linear transforms (as is standard for spectral decompositions), the underlying subspace and its behavioral interpretation are stable, reproducible and identifiable.

**Summary.**   MCD inherits the identifiability properties of spectral RL methods: the motif basis is identifiable up to an invertible transformation, and the motif *subspace*—the structure that determines $P$, $Q$, and $\pi$—is uniquely recoverable. This stands in contrast to latent-variable models such as HMMs or SLDS, which in general do not admit identifiable latent subspaces without strong, often unrealistic, constraints.

## I  LONG-TERM DEPENDENCY DISCUSSION

Compared with prior approaches, our framework implicitly captures the long-term temporal structure of behavior through the Bellman equation in the reinforcement learning (RL) formulation. The component responsible for encoding long-term and multi-scale dynamics is the motif weight $u(t)$. From the linear decomposition of the state–action value function (Eq. 3),

$$u(t) = w + \gamma \int V(s')\, \mu(s')\, q(s')\, ds',$$

it follows that $u(t)$ integrates two sources of information simultaneously: (i) long-horizon structure through the value function $V(s')$, which summarizes discounted future trajectories, and (ii) short-horizon structure through the immediate reward parameter $w$. Since RL theoretically operates over an infinite horizon, this formulation naturally enables modeling of extended temporal dependencies. Empirically, we observe this structure in Fig.4B: some motifs exhibit stable, slowly varying activation profiles (e.g., motif 1, gray curve), whereas others fluctuate rapidly and capture short-term transitions (e.g., motif 2, pink curve). Intuitively, consider a mouse that intermittently sniffs throughout the day but engages in fast running only during a brief morning period. The coefficient $u_i(t)$ for sniffing would rise gradually across long time windows, while the coefficient $u_j(t)$ for running would increase only during short, specific intervals.

A classical model that also captures temporal dependencies is MotionMapper (Berman et al., 2014), which employs a wavelet transform to decompose motion trajectories in the frequency domain. Although both MotionMapper and MCD may be viewed as spectral methods, they rely on fundamentally different basis functions. Wavelet bases explicitly correspond to specific temporal frequencies, thus providing an inherent multi-scale temporal interpretation. Likewise, classical eigendecomposition of linear dynamical systems yields eigenmodes whose associated eigenvalues determine characteristic relaxation time-scales. For example, if $P = V\Lambda V^{-1}$, then

$$x_t = P^t x_0 = V\Lambda^t V^{-1} x_0,$$

and long-term behavior is dominated by the eigenvector corresponding to the largest eigenvalue. In contrast, our method is equivalent to performing a singular value decomposition (SVD) of the transition operator (Ren et al., 2023), and SVD basis vectors do *not* encode distinct time scales in general. Thus, the motifs $\phi$ in MCD are not required to exhibit temporal-frequency separation. Instead, multi-scale behavioral structure arises from the RL objective itself and is manifested in the evolution of the motif weights $u(t)$. Another conceptual distinction is that MotionMapper captures multi-scale structure in the *frequency domain* of motion, whereas our method captures such structure in the *policy domain* via a generative decision-making model, emphasizing the underlying motivational processes shaping behavior.

## J HYPERPARAMETER DISCUSSION

### J.1 MOTIF DIMENSION

To evaluate the sensitivity of our model to the dimension of the hidden state, we change the dimensions of motifs $d$ and verify them in different datasets.

In the discrete case, there exists a minimum dimension that could fully represent the motif space. In fact, the motif discovery (Eq. 2) is equivalent to SVD of the transition matrix $P(s'|s,a)$(Ren et al., 2023). Since the rank of the transition matrix is $|\mathcal{S}|$, the ranks of $\phi$ and $\mu$ matrices are both $|\mathcal{S}|$, which is exactly the minimum dimension of $\phi(s,a)$ that could fully represent the motif space. This could be clearly seen in the gridworld dataset (Fig. 8) and labyrinth navigation dataset (Fig. 9).

For the gridworld dataset, $d_{min} = 3 * 3 = 9$. If $d < 9$, the SVD could only provide a low-rank approximation. As a result, the recovered reward map is blurred and incorrect (Fig. 8A-C) and the correlation remains low (Fig. 8G). For $d \geq 9$, the motif matrix is full-rank. So the reward maps are nearly perfect (Fig. 8D-F), and the correlations are near 100% (Fig. 8G). Similar analysis also applies to labyrinth navigation dataset where $d_{min} = 127$(Fig. 9). In this case, variant with fewer motif dimensions ($d = 64$) cannot generate realistic reward maps while variants with more motif dimensions ($d \geq 127$) can faithfully recover the reward peaks in each task.

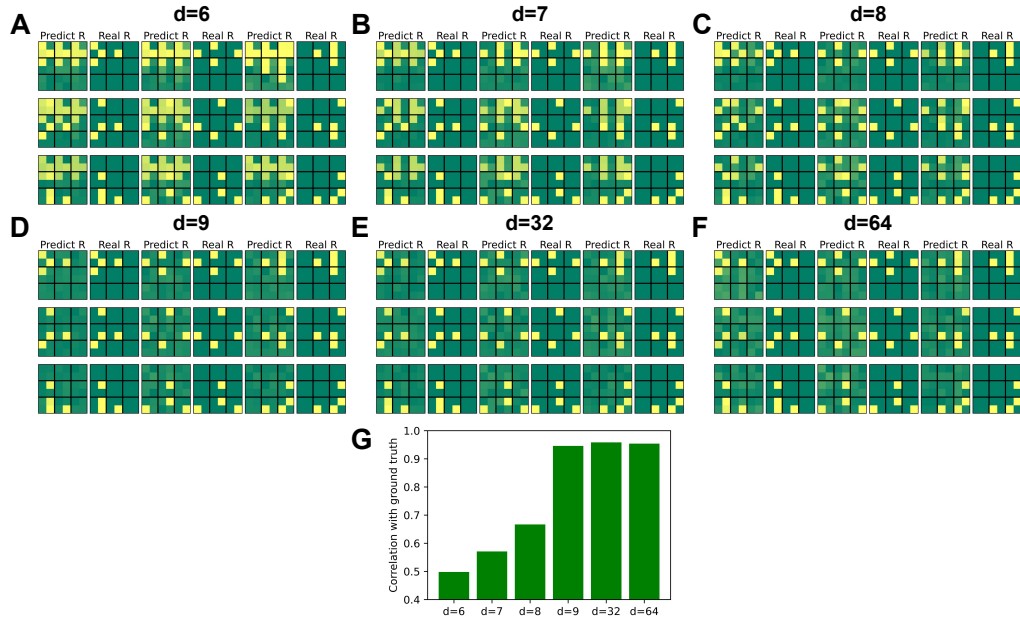

Figure 8: Reward maps generated by models of different motif dimension $d$ in the gridworld dataset.

In the continuous case, since the transition kernel is infinite-dimension, any finite $d$ would only provide a low-rank approximation to the transition kernel. As $d$ increases, the fitting performance would be better. However, since the importance of different motifs is different, by observing the performance under different $d$, we could select one to be as small as possible while still capturing the entire motif space as comprehensively as possible, and maintain the performance. In this experiment (Fig. 10), we select $d = 64$.

### J.2 NOISE DISTRIBUTION

The continuous version of MCD is based on contrastive learning. Therefore, a high-quality negative sample distribution is crucial for the motif and policy learning. Here we evaluate different selections of noise distributions in the continuous free-moving animal behavior dataset.

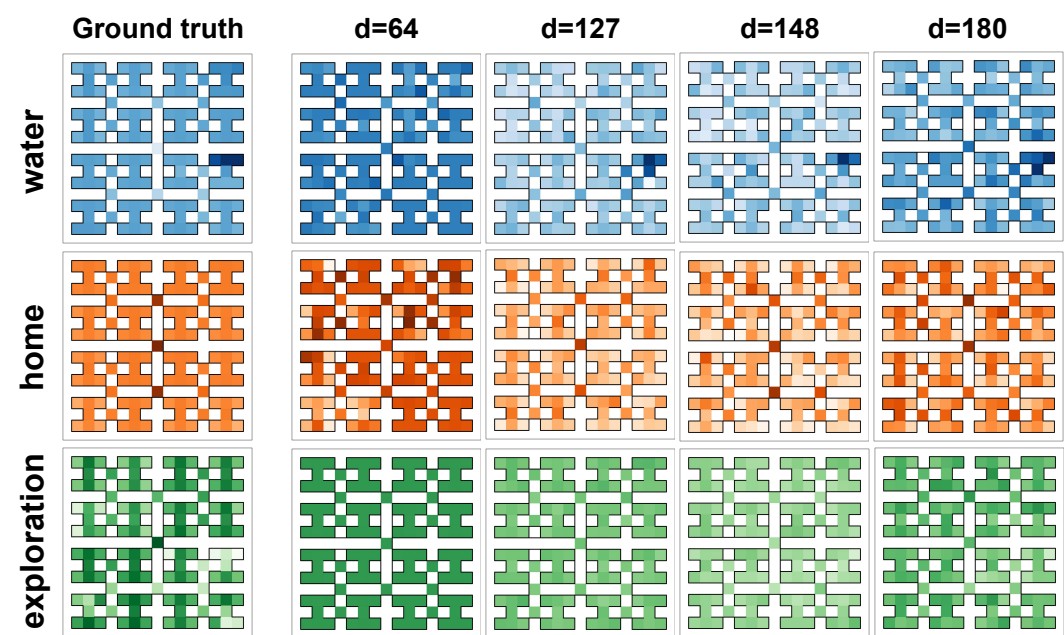

Figure 9: Reward maps generated by models of different motif dimension $d$ in the labyrinth navigation dataset.

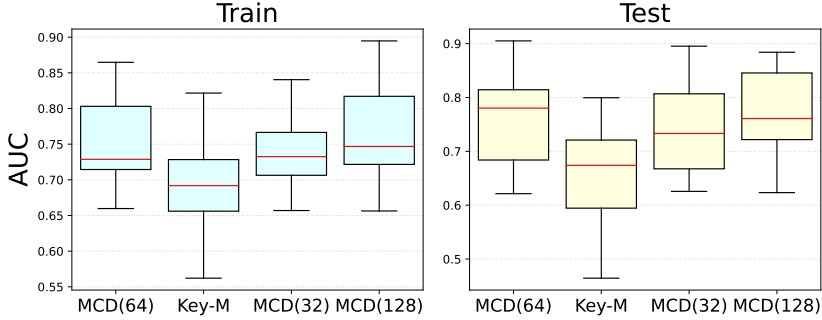

Figure 10: AUC generated by models of different motif dimension $d$ in the animal free-moving dataset.

For the motif learning (Eq. 7), in the main text, we directly sample states from the dataset. In other words, the noise distribution is $\rho(s) = \int \tau^e(s, a)da$.

In "Motif-N" variant, we replace it by a uniform distribution, whose dimension-wise bounds are determined as the maximum and minimum values of the states in the dataset, i.e. $s_i' \sim U(\min(\rho(s_i)), \max(\rho(s_i)))$. Because the negative samples are not good enough, the resulting energy-based model can neither capture high-quality motifs nor provide good basis vectors for later policy learning. Therefore the AUC score remains low (Fig 11).

For the policy learning (Eq. 8), in the main text, we also sample negative samples directly from the dataset $(s', a') \sim \tau^e$ and use $a'$ as a negative sample of actions. In other words, the noise distribution is $\zeta(a) = \int \tau^e(s, a)ds$.

In the "Policy-N" variant, we sample negative pairs from uniform distribution $a_i' \sim U(\min(\zeta(a_i)), \max(\zeta(a_i)))$, similar to "Motif-N". In the "AR-Actor" variant, we additionally train an auxilliary autoregressive actor $\pi(a|s)$ to fit $\pi^e(a|s)$ using MLE. And the negative samples are from this actor $\pi(a|s)$. For each dimension, the actor has a network of two hidden layers, with hidden dimension=256 each, to produce the mean and variance of this action dimension, i.e. $a_i \sim \mathcal{N}(\mu(a_i|s, a_{<i}), \sigma^2(a_i|s, a_{<i}))$. As expected, uniform noise distribution is not good enough, and "Policy-N" achieves lower AUC than the original version. And the "AR-Actor" achieves similar

performance on the test dataset, proving the robustness of MCD to noise distribution if it is good enough.

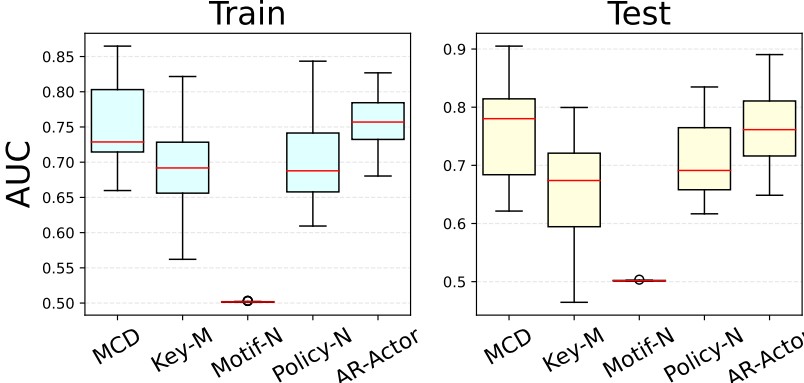

Figure 11: AUC generated by models trained under different noise distribution in the animal free-moving dataset.

We also tested the influences of the number of negative samples (Fig. 12). MCD is consistently better than Key-Moseq and maintains a stable performance. It turns out that the model is robust against the choice of number of negative samples.

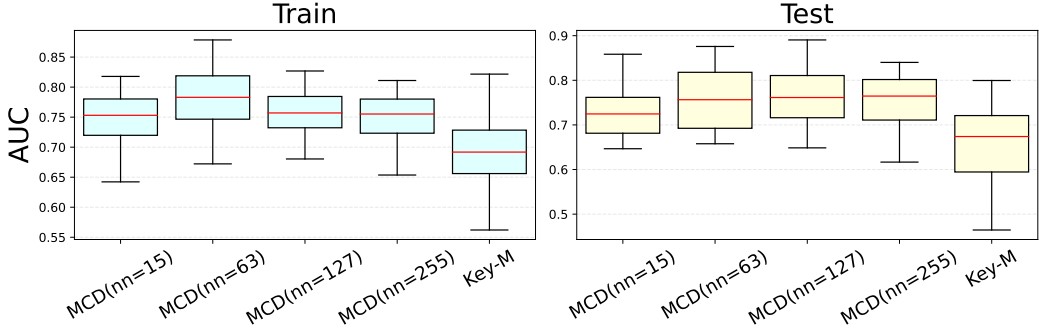

Figure 12: AUC generated by models trained using different numbers of negative samples for motif contrastive learning in the animal free-moving dataset. $nn = k$ means the **n**umber of **n**egative samples per positive sample is $k$.

### J.3 DATASET PARTITION

In the third experiment, we randomly split the whole dataset into training set and test set. Here, we train the model on some mice and then test it on other heldout mice, i.e. split the dataset by mouse identities. This experiment shows whether the motifs learned by the model are general and transferrable across different subjects. The result is shown in Fig. 13 where the complete exclusion of one animal from the training set does not impair the performance. This experiment proves that the MCD can learn a general set of motifs that could be transferred across animals.

### J.4 GAUSSIAN RANDOM WALK PRIOR

In the free-moving animal behavior dataset (Sec. 4.3), we impose a Gaussian random-walk prior on the time-varying motif weights $u(t)$ to encourage temporal smoothness. To assess the sensitivity of our method to this regularization, we varied the strength of the Gaussian prior and re-evaluated model performance (Fig. 14). As expected, very large coefficients introduce slight degradation due to oversmoothing. However, across all tested values, performance remains substantially higher than

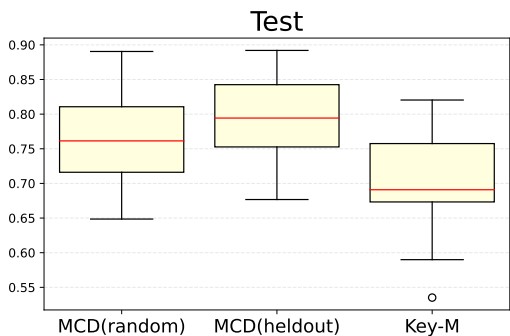

Figure 13: AUC generated by models trained using different train-test split methods. For MCD(random), the dataset is randomly split. For MCD(heldout), the behavior of one mouse is specially held out as the test set while the rest is taken as the training set.

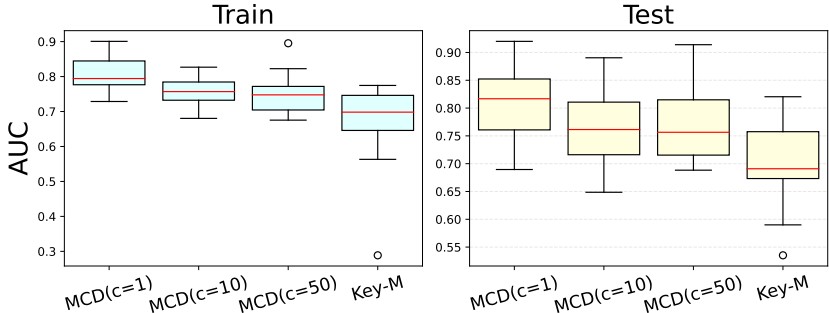

Figure 14: AUC generated by models tested using different Gaussian random walk prior coefficients in the animal free-moving dataset.

the baseline model. This demonstrates that MCD is robust to the choice of Gaussian random-walk regularization strength and does not rely on fine-tuning this hyperparameter to achieve strong results.

## K  SCIENTIFIC IMPACT AND RELATIONS TO NEUROSCIENCE AND ETHOLOGY

**Scientific relevance to neuroscience.**  Our work is primarily motivated by the neuroscience perspective, where the goal is to extract interpretable behavioral structure that can be directly linked to neural circuit dynamics, internal motivational states, and decision-making processes. From this viewpoint, MCD provides scientifically meaningful variables: it discovers low-level motor motifs that correspond to reproducible movement primitives and models behavior as smoothly varying mixtures of these primitives. This mirrors known neural control principles in the motor cortex, basal ganglia, and brainstem, where overlapping action components, not discrete switches, combine to generate natural movement. The resulting motif representations and time-varying policy weights offer a rich, biologically interpretable representation for analyzing how neural populations evolve alongside behavior.

The learned representations can be mapped to neural representations in future work. Each motif $\phi(s, a)$ defines a low-level motor primitive, giving a clear behavioral regressor for examining whether neurons in *motor cortex* encode specific movement components. The time-varying policy weights $w(t)$ and reward-related weights $u(t)$ describe how motifs are combined and modulated over time, offering hypotheses about potential control- and value-related signals in the *dorsal striatum*. While we do not perform neural analyses in this paper, the structured motif representation provides a principled framework for relating behavior to activity in these circuits in future neuroscience studies.

For the training protocol, our method is trained purely from behavior; no neural signals, reward labels, or joint behavior–neural objectives are used. This distinguishes our approach from models that rely on neural data to infer latent goals or value functions. On the other hand, although our method does not require neural recordings, the learned motifs correspond to low-dimensional dynamical components of behavior that can in principle be aligned with neural manifolds (e.g., cyclic modes for gait, motor primitives, or population attractors).

**Relevance to ethology and real-world behavior.**  Beyond neuroscience, MCD is well suited for ethological studies where the goal is to characterize behavior in naturalistic, minimally constrained environments across multiple timescales. By capturing long-range dependencies, allowing motifs to co-occur, and modeling continuous dynamics, MCD can describe multi-scale organization of behavior, such as exploratory sequences, foraging patterns, or grooming hierarchies, without assuming discrete states. This enables ethologists to quantify how natural behaviors are composed, how they transition, and how they evolve over long durations. Thus, while our emphasis is on neuroscience applications, the method is fully compatible with ethological framework and supports general scientific questions about the structure and function of natural behavior.

## L    REWARD MAPS OF LABYRINTH NAVIGATION DATASET INFERRED BY DIFFERENT MODELS

MCD's result of the navigation labyrinth dataset is based on the task segmentation of Ke et al. (2025). There remains two questions: (1) how do other models perform on the same segmentation; and (2) whether the ground truth from Ke et al. (2025) is valid. To further validate our results, we evaluated two additional inverse reinforcement learning baselines: (1) IQ-Learn (IQL, Garg et al. (2021)), where we use the same segmentation from Ke et al. (2025) to infer a reward map for each task; and (2) Dynamic-IRL (DIRL, Ashwood et al. (2022)), where without relying on this segmentation, we infer a time-varying mixture of reward functions directly from their curated data. We use a tabular version of IQL here just like MCD. For comparison, we constructed two auxiliary reward maps: a perfect reward map with artificially placed rewards, and a random baseline map whose entries are sampled from a standard normal distribution. In the perfect map, the water-seeking task reward function has a peak of 1 at the water port. The home-seeking task reward function has a peak of 1 at the home location. And the exploration task reward function has a uniform low reward of 0.3 at every location.

Fig. 15A shows all inferred maps, and Fig. 15B reports the Pearson correlation coefficients across all maps. There are three important conclusions. (1) The strong alignment between the MCD-recovered maps, the ground-truth maps from Ke et al. (2025) (GT), and the perfect manual map demonstrates that MCD successfully recovers the underlying motivational structure. (2) MCD achieves consistently higher correlation with both GT and the perfect map than IQL under the same task segmentation, suggesting that MCD captures the animal's latent motivation more accurately than the IQL baseline. (3) The high correlation between GT, the perfect map, and DIRL provides external validation for the quality of the GT annotations themselves.

Note that although DIRL achieves higher correlations with the ground-truth maps, it is not suitable for our setting. DIRL requires stereotyped, repeated sequences of *identical* length and therefore depends on carefully curated datasets in which such fixed-length segments can be extracted (not the same dataset as we use, but the same environment). In contrast, our datasets—especially the continuous animal free-moving dataset—exhibit substantial heterogeneity and do not contain repeated trajectories of uniform duration. We are also concerned that such curation would introduce bias and distort the natural statistics of the behavior. Furthermore, DIRL relies on *tabular* value iteration to backpropagate gradients from the policy to the reward, which is fundamentally incompatible with the continuous state and action spaces present in the animal free-moving behavior data.

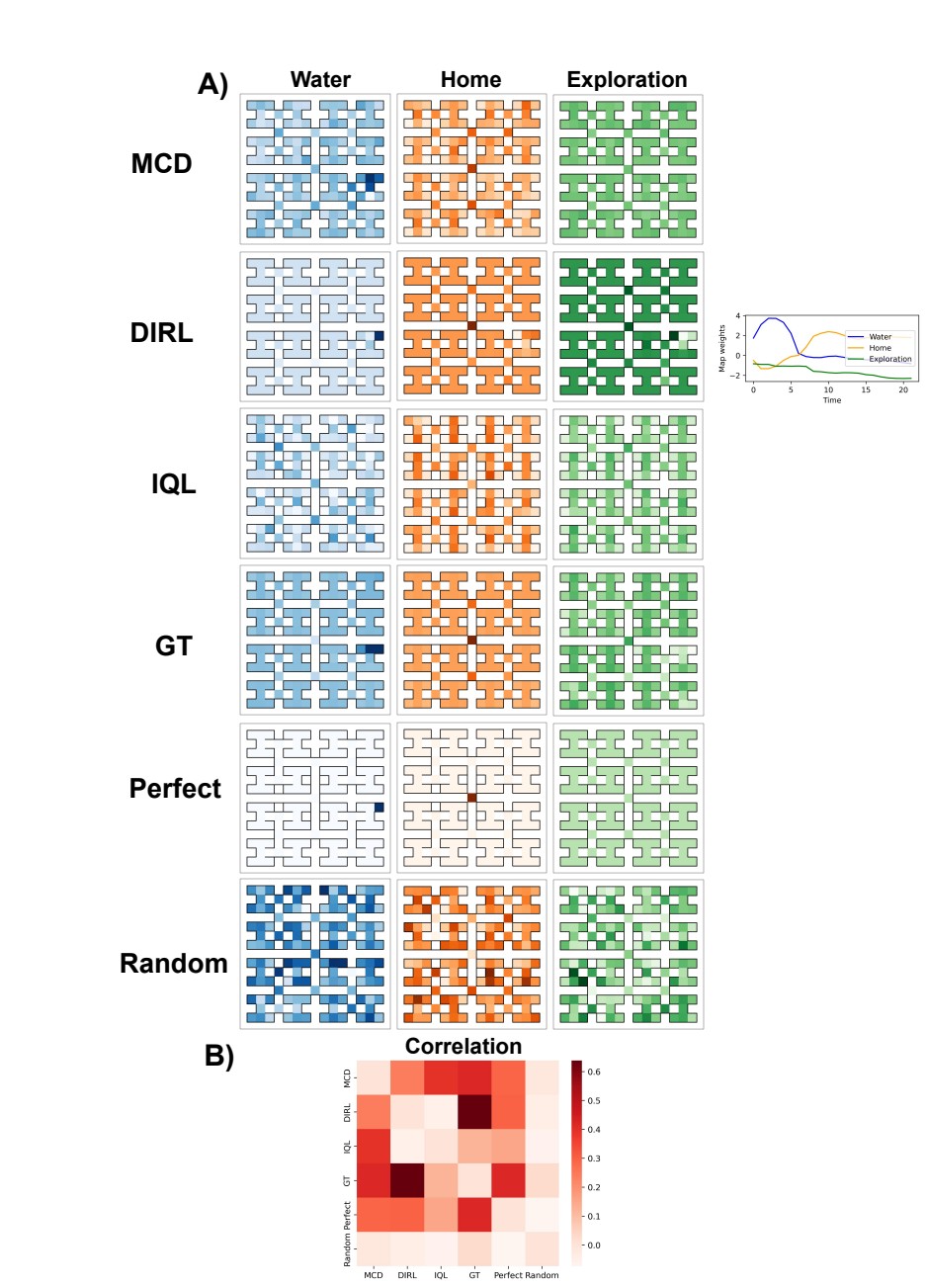

Figure 15: A. Reward maps inferred by different models. The maps from DIRL are ranked based on the time-varying weights (right at the corresponding row). "Water" weight increases and then decreases; "home" weight remains low and then increases; and "exploration" weight remains low. **B.** Correlations across the reward maps.

## M CONNECTIONS TO STATE-SPACE MODEL

In the animal free-moving behavior dataset, our comparison includes both discrete segmentation models (e.g., Key-MoSeq), which assign behaviors to discrete syllables, and continuous trajectory models (OPAL, SemiSeg), which capture smooth behavioral dynamics without imposing explicit motif boundaries. While our discussion of the discrete models was more detailed, we expand this section to clarify two additional points. First, the continuous baselines we use—specifically the SemiSeg variant—are in fact continuous state-space models that model latent behavioral dynamics through recurrent transitions and stochastic emissions. Second, despite the expressive capacity of these continuous SSMs, our proposed MCD framework achieves markedly stronger performance, indicating its advantage in capturing the fine-grained and compositional structure of animal behavior.

With the latent dynamic state $h_t$, SemiSeg models the sequence $(s_1, \ldots, s_T)$ via a deterministic latent transition model, and a stochastic emission model:

$$h_{t+1} = f_\omega(h_t, s_t), \quad \text{with } p_\omega(h_{t+1} \mid h_t, s_t) = \delta\big(h_{t+1} - f_\omega(h_t, s_t)\big), \tag{17}$$

$$s_{t+1} \sim p_\theta(s_{t+1} \mid h_{t+1}) = \mathcal{N}\big(\mu_\theta(h_{t+1}), \sigma^2 I\big), \tag{18}$$

where $f_\omega$ is a GRU transition and $\mu_\theta$ is a neural network emission function. The delta function can be viewed as the limit of a Gaussian transition with $\sigma \to 0$, corresponding to a deterministic latent transition.

Under this construction, the joint distribution over latent and observed variables can be factorized as

$$p(h_{1:T}, s_{1:T}) = p(h_1) \, p(s_1 \mid h_1) \prod_{t=1}^{T-1} p_\omega(h_{t+1} \mid h_t, s_t) \, p_\theta(s_{t+1} \mid h_{t+1}). \tag{19}$$

This is exactly the state-space factorization: a Markovian latent process $h_{t+1}$ evolving in time, together with a stochastic emission model generating the observed states. The dependence on $s_t$ in the transition corresponds to an input-driven SSM, with the previous state acting as an input.

We train SemiSeg by maximizing the log-likelihood $\log p(s_{1:T})$ via a variational lower bound. Introducing an approximate posterior over the latent trajectory $q_\phi(h_{1:T} \mid s_{1:T})$, the evidence lower bound (ELBO) $\mathcal{L}$ can be acquired through

$$\log p(s_{1:T}) \geq \mathcal{L} = \mathbb{E}_{q_\phi(h_{1:T} \mid s_{1:T})}[\log p(h_{1:T}, s_{1:T}) - \log q_\phi(h_{1:T} \mid s_{1:T})] \tag{20}$$

$$= \mathbb{E}_{q_\phi(h_{1:T} \mid s_{1:T})}[\log p(s_{1:T} \mid h_{1:T})] - \mathrm{KL}\big(q_\phi(h_{1:T} \mid s_{1:T}) \,\|\, p(h_{1:T})\big) \tag{21}$$

It is easy to factorize the first term

$$\log p(s_{1:T} \mid h_{1:T}) = \sum_{t=1}^{T} \log p_\theta(s_t \mid h_t). \tag{22}$$

For the deterministic transition $h_{t+1} = f_\omega(h_t, s_t)$ with $p_\omega(h_{t+1} \mid h_t, s_t) = \delta(h_{t+1} - f_\omega(h_t, s_t))$, a natural variational family is

$$q_\phi(h_{1:T} \mid s_{1:T}) = q_\phi(h_1 \mid s_{1:T}) \prod_{t=1}^{T-1} \delta\big(h_{t+1} - f_\omega(h_t, s_t)\big), \tag{23}$$

i.e. the posterior has freedom only in the initial latent state $h_1$ and shares the deterministic transition with the prior $p(h_{1:T})$. Under this choice, the delta functions cancel in the KL, and the regularization term reduces to

$$\mathrm{KL}\big(q_\phi(h_{1:T} \mid s_{1:T}) \,\|\, p(h_{1:T})\big) = \mathrm{KL}\big(q_\phi(h_1 \mid s_{1:T}) \,\|\, p(h_1)\big), \tag{24}$$

where $p(h_1) = \mathcal{N}(0, I)$ is the initial prior. The ELBO therefore simplifies to

$$\mathcal{L} = \sum_{t=1}^{T} \mathbb{E}_{q_\phi(h_{1:T} \mid s_{1:T})}[\log p_\theta(s_t \mid h_t)] - \mathrm{KL}\big(q_\phi(h_1 \mid s_{1:T}) \,\|\, \mathcal{N}(0, I)\big). \tag{25}$$

For the Gaussian emission model

$$p_\theta(s_t \mid h_t) = \mathcal{N}\big(s_t; \mu_\theta(h_t), \sigma^2 I\big), \tag{26}$$

the negative log-likelihood decomposes (up to an additive constant) as

$$-\log p_\theta(s_t \mid h_t) = \frac{1}{2\sigma^2} \big\| s_t - \mu_\theta(h_t) \big\|_2^2 + \text{const.} \tag{27}$$

Consequently, the training objective (negative ELBO, ignoring constants) is

$$\mathcal{J}(\theta, \omega, \phi) = \frac{1}{2\sigma^2} \sum_{t=1}^{T} \mathbb{E}_{q_\phi(h_{1:T}|s_{1:T})} \big\| s_t - \mu_\theta(h_t) \big\|_2^2 + \mathrm{KL}\big(q_\phi(h_1 \mid s_{1:T}) \,\|\, \mathcal{N}(0, I)\big). \qquad (28)$$

In particular, if we fix $\sigma^2 = 1$ and use a deterministic initial latent (e.g. $h_1 = 0$), the loss reduces to a mean-squared reconstruction term, consistent with SemiSeg (Whiteway et al., 2021).

SemiSeg is therefore a continuous neural state-space model with: (i) a deterministic latent dynamical state updated by a recurrent transition $h_{t+1} = f_\omega(h_t, s_t)$, and (ii) a stochastic emission model $p_\theta(s_{t+1} \mid h_{t+1})$. Given that MCD still performs better than SemiSeg (Fig. 4.3), our model thus demontrates more powerful ability to capture continuous behavior dynamics than continuous state-space models.

# N    GENERATED ROLLOUT TRAJECTORIES

In the animal free-moving dataset, we train a generative energy model to estimate the state–action value function. Here, we assess its reliability using Hamiltonian Monte Carlo (HMC) sampling. For each motif $d_i$, we construct a one-hot vector $u$ with $u_{d_i} = 1$ and $u_{d_j} = 0$ for all $j \neq i$, and then sample an action $a$ from the learned energy model. For each $d_i$, we sample $10$ actions in total. The resulting motif-specific rollout trajectories (Fig 16) largely align with the empirical averages (Fig. 7B), demonstrating that the model captures the dominant behavioral tendencies. A small number of trajectories deviate from the empirical trends, probably because $u$ is never used as a strict one-hot vector during training but instead appears as a mixture over motifs. We attribute these deviations to unavoidable generative noise and the distributional mismatch between test-time one-hot inputs and the mixed representations observed in real data.

For that, we choose several clips and generate the rollouts (number of rollout steps=5) based on the inferred $u(t)$. The alignment between generated trajectories and real trajectories (Fig. 17) show that MCD could capture the real behavior dynamics.

**HMC parameters.**    We use a step size of $1 \times 10^{-4}$, 100 leapfrog steps per action, temperature $= 1$, mass matrix $M = I$, and initial momentum $r \sim \mathcal{N}(0, 1/200)$. The leapfrog integrator is employed to improve sampling accuracy.

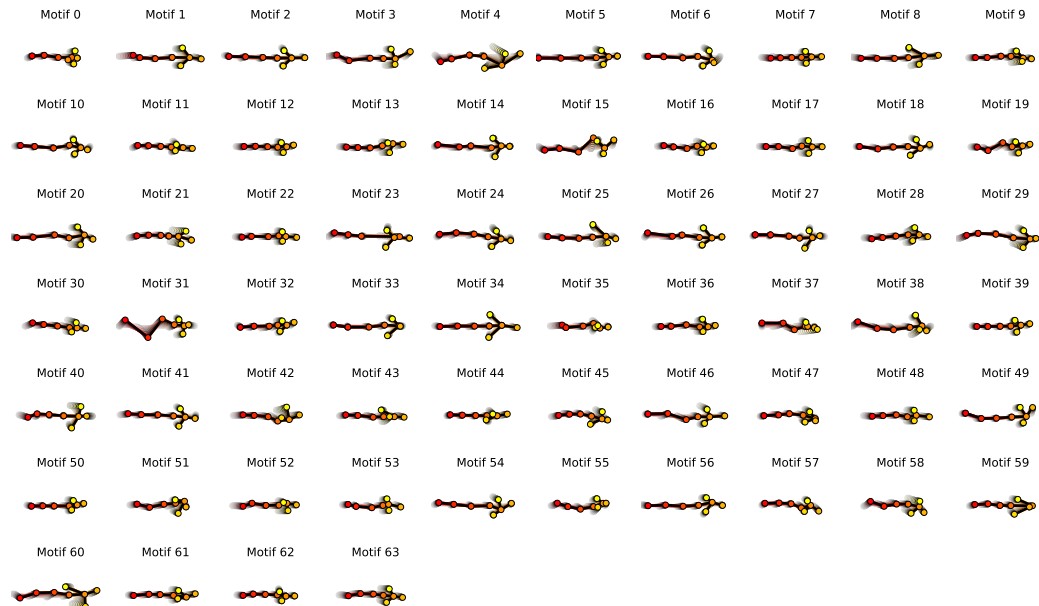

Figure 16: Motif-specific rollout trajectories generated by HMC sampling from the trained energy model. Indexes are consistent with Fig. 7.

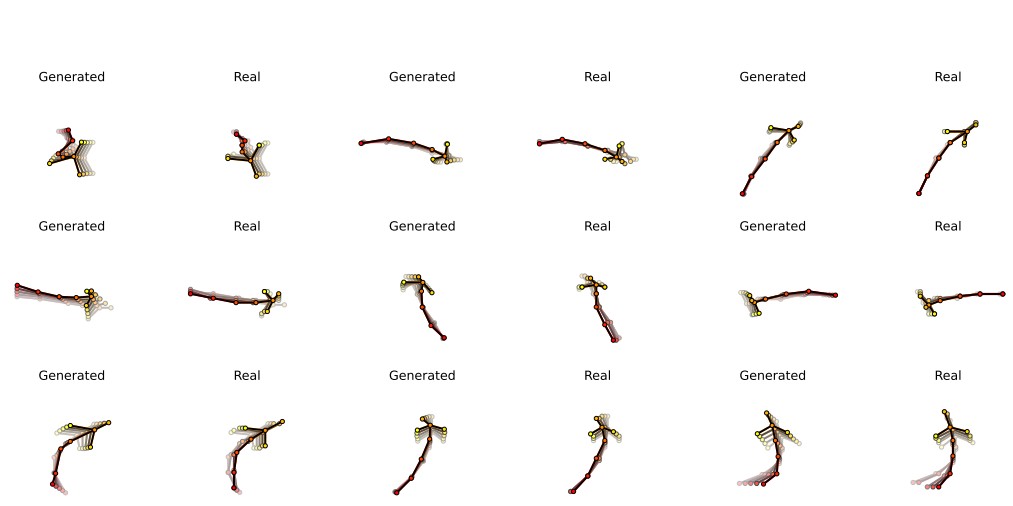

Figure 17: Rollout trajectories based on inferred $u(t)$ generated by HMC sampling from the trained energy model.

