# OpenReview forum: "Learning Task-Agnostic Motifs to Capture the Continuous Nature of Animal Behavior"
_ICLR.cc/2026/Conference — Submitted to ICLR 2026_

### Official Review · Reviewer_9p6R · 2025-10-30

**Soundness:** 2
**Presentation:** 2
**Contribution:** 3
**Rating:** 4
**Confidence:** 3

**Summary:**

This paper proposes a task-agnostic motif framework that models animal behavior as a continuous, compositional mixture rather than discrete syllables. The method learns low-dimensional motif bases and expresses reward, value, and policy on the same basis, enabling soft blending of behaviors. For continuous spaces, transitions are modeled with an energy-based approach and trained via noise-contrastive estimation, including a learned mapping from internal features to motifs. In the experiments, synthetic gridworld (recovery of known rewards), real maze navigation (agreement with Ke et al.’s reward maps used as a proxy ground truth), and free behavior (AUC-based discrimination and qualitative visualizations) were examined, where the proposed method outperforms Keypoint-MoSeq, SemiSeg, and OPAL on the reported metric in the last experiment.

**Strengths:**

Originality lies in shifting from discrete segmentation to a continuous, compositional description that unifies behavior interpretation and policy learning. The technical design is coherent across settings, with clear explanations and visualizations that make the idea of blending motifs intuitive. The evaluation spans synthetic, controlled, and naturalistic data, showing consistent advantages on the chosen metric. The concept of a reusable, low-dimensional basis that both explains behavior and supports control is potentially impactful for neuroscience and IL/RL applications.

**Weaknesses:**

The manuscript may lack sufficient methodological detail about the NCE negative-sample distribution (ρ), and it may not clearly position the motif idea relative to related robotics primitives or recent RL/IL behavioral-modeling work.  Evaluation may be also limited in the second and third experiments. The details are given in the following questions.

**Questions:**

1. The concrete definition and implementation of the noise distribution ρ (i.e., from which distribution you sample the negatives) are not fully clear to me. For robustness and reproducibility, could you specify exactly how ρ is chosen and implemented in each experiment? In the future, providing an ablation showing how different choices of ρ (and the number of negatives) affect training stability and final performance would be better.

2. This study’s motifs concept appears very close to robotics literature on primitives/skills with mixing or blending. While some citations are given, I feel a more systematic positioning in Related Work would be helpful. For example, is it possible to compare (a) the basis relation (e.g., linear combination of probabilistic movement primitives vs. linear reward/Q on motifs in this study, including latent-space structure and regularization), (b) learning regimes (supervised / IL / RL / offline RL), and (c) treatment of normalization (need for EBM/NCE)?

3. Recent works that explain animal behavior via RL/IL are cited, but the background positioning is also not fully clear to me. To make your novelty explicit, is it possible to clarify: (a) the objective functions (MaxEnt, IRL/IL variants), (b) learning signals used (behavior-only vs. joint optimization with neural data), and (c) whether and how your motif space can be related or mapped to neural representations?

4. In the second experiment, When using the Ke et al. reward maps as a valid ground truth, is it possible to provide cross-validation of external validity? For example, agreement with other algorithms or with human annotations (e.g., correlation, map distance metrics, overlap of high-reward zones)?

5. In the third experiment, in addition to AUC, is it possible to evaluate long-horizon rollouts from the learned policy against statistics of real data, and/or a blinded human discrimination test (human judges comparing generated vs. real trajectories)? Either would provide complementary evidence for generative quality and interpretability.

---

> ### Author Response · Authors · 2025-11-21
> **Part 1**
>
> **Question 1**
>
> For the motif learning, $\rho(s)=\int \tau^e(s,a)da$ is the marginal state distribution of $\tau^e$ (state-action distributions generated from the dataset, Eq. 7). $\rho$ is only used once here. We have clarified this in the main text. Another place involves NCE is when you want to learn $\pi(a|s)$ in the continuous version (Eq. 8). Here the noise distribution is the data distribution $\tau^e$. We have included the discussions of how different choices of noise distribution affect the final performance. We will post other results about the training stability and how the number of negatives affect the results in the next few days.
>
> **Question 2**
>
> Thanks for the questions.
>
> (a) SkillBlender[1] uses linear combinations of lower-level controller outputs as the final actions $a_t=\sum_i k_ia^i_t$. While we use the linear combinations of different motifs to generate the final state-value functions $Q_t=\sum_i \phi_i(s,a)u^i_t$ as well as the transition kernel $P(s'|s,a)=\sum_i \phi_i(s,a)\mu_i(s')$. MCD has two advantages: (1) This shared motif in MCD is more fundamental and interpretable in revealing the basic structure of the animal's intentions. (2) A naive mixture of the low-level controller is less biologically realistic in modeling animal behavior than mixtures of state-value functions. The latter can find supporting evidence in neuroscientific literature[2]. We have discussed other literature in linear structure of environment/policy in Appendix G.
>
> (b) Discussions on offline imitation learning, motif/skill-based RL has been added to Appendix G.
>
> (c) For $\nu$ and $\psi$, we normalize the representation $\psi$ and $\nu$ to sum-1 to avoid the identifiability problem up to scaling. Otherwise there does exist a problem of identifiability as well as trainability for all contrastive learning problem: if we just train under a trivial objective function without any constraints, the model would reduce the contrastive loss by simply increasing embedding magnitudes, rather than reducing the angles between the embeddings. And any solution could be scaled by rescaling $\psi’ =s\psi, \nu’ =\frac{1}{s}\nu$. Normalization would solve this problem. This normalization technique is widely adopted in contrastive learning literature. We will add the comparisons with other techniques in the next few days.
>
> **Question 3**
>
> (a) Novelty statement.
>
> (1) For the policy learning of the discrete version, our maximal-likelihood objective is similar to MaxEnt IRL, but we do not infer rewards as necessary intermediate steps, although they can be inferred a posterior.
>
> (2) For the policy learning of the continuous version, there are several other novelty points. First, we adopts NCE objective that factorizes the policy into a time-independent motif basis $\phi(s,a)$ and time-varying weights $u(t)$, but this clear linear policy structure that separates motifs and latent variables is absent in previous motif discovery literature (Appendix G).
>
> (3) For the environment assumption, we have minimal assumption for the environment structure. For example, we do not assume the behavior dynamic is linear (like $s_t=As_{t-1}$, as in hidden-Markov-model-based behavior segmentation works).
>
> (b) Training data source.
>
> Our method is trained purely from behavior; no neural signals, reward labels, or joint behavior–neural objectives are used. This distinguishes our approach from models that rely on neural data to infer latent goals or value functions.
>
> On the other hand, although our method does not require neural recordings, the learned motifs correspond to low-dimensional dynamical components of behavior that can in principle be aligned with neural manifolds (e.g., cyclic modes for gait, motor primitives, or population attractors).
>
> (c) Relations to neural representations.
>
> Our motif space is designed so that possible neural correlates can be explicitly tested in future work. Each motif $\phi(s,a)$ defines a low-level motor primitive, giving a clear behavioral regressor for examining whether neurons in *motor cortex* encode specific movement components. The time-varying policy weights $w(t)$ and reward-related weights $u(t)$ describe how motifs are combined and modulated over time, offering hypotheses about potential control- and value-related signals in the *dorsal striatum*. While we do not perform neural analyses in this paper, the structured motif representation provides a principled framework for relating behavior to activity in these circuits in future neuroscience studies.
>
> We have included these discussions to Appendix K.
>
> [1] Kuang, Y., ... & Wang, Y. (2025). SkillBlender: Towards Versatile Humanoid Whole-Body Loco-Manipulation via Skill Blending. arXiv preprint arXiv:2506.09366.
>
> [2] Makino, H. (2023). Arithmetic value representation for hierarchical behavior composition. Nature neuroscience, 26(1), 140-149.

---

> ### Author Response · Authors · 2025-11-21
> **Part 2**
>
> **Question 4**
>
> Thanks for the question concerning cross-validation. We have provided the external validation in Appendix L. We included another IRL baseline (DIRL) to show the reasonability of the ground truth using the correlation metric. Beyond that, we also included another baseline (IQL) to show MCD's performance advantage under the same task segmentation. Thanks for taking time to check them.
>
> **Question 5**
>
> We thank the reviewer for the thoughtful suggestion to evaluate long-horizon rollouts as complementary measures of generative quality. While our main focus in Experiment 3 is evaluating how well the learned value function supports accurate motif inference (AUC), we agree that examining the generative behavior of the learned policy is informative. To this end, we have already conducted an additional analysis based on *Hamiltonian Monte Carlo* (HMC) rollouts from our learned energy-based model in Appendix N. We also include a gif in the supplementary material for better visualization. Besides, we have already added an additional expert-label alignment experiment as complementary evidence to support the performance advantage of MCD.(See Section 4.4)

---

> > ### Author Response · Authors · 2025-12-04
> > **Adding comments**
> >
> > We added additional responses to the comments several days ago. Thank you for taking the time to check them.

---

### Official Review · Reviewer_3HVh · 2025-10-31

**Soundness:** 3
**Presentation:** 3
**Contribution:** 3
**Rating:** 6
**Confidence:** 2

**Summary:**

The paper introduces Motif-based Continuous Dynamics (MCD) for modeling animal behavior as continuous compositions of a finite set of reusable motor motifs. Methodologically, MCD (i) learns interpretable motif “bases” by exploiting representations of behavioral transition structure, and (ii) models time-varying behavior as smoothly evolving mixtures of those motifs. Experiments span a multi-task gridworld, a labyrinth navigation task, and freely moving animal behavior, showing reusable components, realistic trajectory generation, and advantages over discrete-segmentation approaches.

**Strengths:**

1. Recasts behavior modeling as continuous mixtures of interpretable motifs, moving beyond discrete syllables.
2. Multi-environment evaluation suggests motif reusability and improved generative realism compared to discrete segmentation.
3. Offers an interpretable generative account of behavior that could aid cross-task generalization, analysis of natural behavior, and links to neural data.

**Weaknesses:**

1. Compare not only to discrete segmentation (HDP-HMM/AR-HMM, SLDS, MoSeq-style) but also modern continuous SSMs (e.g., Neural SSM/RSSM, N-SLDS) that capture smooth trajectories without explicit motifs.
2. Demonstrate cross-task/subject transfer: learn motifs on subset A, evaluate reuse and performance on held-out tasks/animals (quantify similarity up to permutation/rotation).
3. Beyond visuals, report held-out log-likelihood, forecasting error, and realism/coverage metrics; if annotations exist, quantify alignment (e.g., NMI/ARI, MI).

**Questions:**

see weakness

---

> ### Author Response · Authors · 2025-11-21
> **Thanks for the reply**
>
> **Weaknesses 1**
>
> We thank the reviewer for the helpful suggestion to include modern continuous state-space models (SSMs), such as Neural SSMs, RSSMs, and nonlinear SLDS variants. We would like to clarify that our continuous baselines already include such a model. In particular, **SemiSeg** is a **continuous state-space model**. It introduces a latent dynamical state $h_t$ and models behavior using
>
> (i) a deterministic recurrent latent transition $h_{t+1} = f_\omega(h_t, s_t),$ parameterized by a GRU, and
>
> (ii) a stochastic emission model $p_\theta(s_{t+1}\mid h_{t+1}) = \mathcal N(\mu_\theta(h_{t+1}), \sigma^2 I).$
>
> This yields the standard SSM factorization: $
> p(h_{1:T}, s_{1:T}) = p(h_1)p(s_1\mid h_1) \prod_{t=1}^{T-1} p_\omega(h_{t+1}\mid h_t, s_t)p_\theta(s_{t+1}\mid h_{t+1}),$
>
> which matches the generative structure used in the continuous SSMs referenced by the reviewer.
>
> We would like to emphasize that our proposed **MCD** method outperforms both
>
> (i) *discrete segmentation models* (e.g., Key-MoSeq) and
>
> (ii) *continuous SSM-style models* (SemiSeg SSM),
>
> demonstrating that MCD captures both the smoothness of naturalistic behavior and the compositional structure that existing discrete and continuous approaches fail to recover. We appreciate the reviewer’s suggestion and have incorporated these more detailed derivations and clarifications into the revised manuscript (Appendix M).
>
> **Weaknesses 2**
>
> Thanks. We have provided the held-out test result in Appendix J.3. You could also see discussions of other hyperparameters  in Appendix J.
>
> **Weaknesses 3**
>
> Thanks for the suggestion. In fact, we have reported an AUC metric which measures how well the model can distinguish positive samples from negative samples (Figure 4A). And we have further clarified this in the response to reviewer wNqB. Furthermore, we agree that evaluation on real human labels is important. For that, we included expert labels for animal free-moving behavior dataset and report the alignment score in Section 4.4.

---

> > ### Author Response · Authors · 2025-12-04
> > **Adding new rebuttals**
> >
> > We added additional responses to the comments several days ago. Thank you for taking the time to check them.

---

### Official Review · Reviewer_5JMq · 2025-10-31

**Soundness:** 3
**Presentation:** 3
**Contribution:** 2
**Rating:** 6
**Confidence:** 2

**Summary:**

A method for extracting "motifs" (a set of elemental behaviors) from animal behavior trajectories is proposed. The method is based on a modal decomposition of the transition kernel of the environment and a reward function modeled using such modes. In the paper, they show that the maximum entropy policy based on the action-value function can also be written in terms of the modes. They then suggest a method to estimate such modes (and their activations) using the noise contrastive estimation.

**Strengths:**

The method is principled. It is nicely derived based on the MDP with minimal assumptions.

The analyses of the experimental results are in-depth and detailed.

**Weaknesses:**

Although the current results are nicely analyzed, the applicability to various real-world data is not very clear.

Discussion on the method's utility from the ethological or neuroscience points of view is missing. We may be able to read and somehow "interpret" the results, but it is not clear how these are scientifically meaningful. This lack of a domain expert's analysis might make the argument sound slightly arbitrary.

**Questions:**

**(1)**
The method is still not completely clear. Is there any constraint on the scales of the estimated quantities or functions, $\psi$, $\nu$, $f$, $u$? If they are not constrained, some of them seem to be unidentifiable.

**(2)**
The authors contrast the proposed method with "dynamics-based" approaches. I am not quite sure of the intention here. To me the proposed method looks a kind of dynamics-based too, because it is based on the estimation of the quantities related to the dynamics (the policy can also be seen as a part of the dynamics $s \to s'$). This is a question just out of curiosity and is not a fatal factor in my evaluation though.

---

> ### Author Response · Authors · 2025-11-21
> **Thanks for the reply**
>
> **Weaknesses**
>
> Our work is primarily motivated by the neuroscience perspective, where the goal is to extract interpretable behavioral structure that can be directly linked to neural circuit dynamics, internal motivational states, and decision-making processes. From this viewpoint, MCD provides scientifically meaningful variables: it discovers low-level motor motifs that correspond to reproducible movement primitives and models behavior as smoothly varying mixtures of these primitives. This mirrors known neural control principles in the motor cortex, basal ganglia, and brainstem, where overlapping action components, not discrete switches, combine to generate natural movement. The resulting continuous motif activations and time-varying policy weights offer a rich, biologically interpretable representation for analyzing how neural populations evolve alongside behavior.
>
> At the same time, the method is not limited to neural datasets. MCD naturally extends to broader ethological studies, where the goal is to characterize behavior in unconstrained, naturalistic environments across multiple timescales. By capturing long-range dependencies, allowing motifs to co-occur, and modeling continuous dynamics, MCD can describe multi-scale organization of behavior, such as exploratory sequences, foraging patterns, or grooming hierarchies, without assuming discrete states. This enables ethologists to quantify how natural behaviors are composed, how they transition, and how they evolve over long durations. Thus, while our emphasis is on neuroscience applications, the method is fully compatible with ethological framework and supports general scientific questions about the structure and function of natural behavior.
>
> Due to space limits in the main paper, we added Appendix K to highlight these scientific impacts. We also incorporated expert human labels for a subset of the free-moving dataset and compared them with our model’s predicted behavioral segments and those from other methods. The close alignment between our discovered motifs and expert annotations shows that MCD produces behavior segmentations consistent with domain-expert insight.
>
> **Question 1**
>
> We put the discussion of the identifiability at Appendix H.
>
> For $\nu$ and $\psi$, we normalize the representation $\psi$ and $\nu$ to sum-1 to avoid the identifiability problem up to scaling. Otherwise there does exist a problem of identifiability as well as trainability for all contrastive learning problem: if we just train under the trivial objective function without any constraints, the model would reduce the contrastive loss by simply increasing embedding magnitudes, rather than reducing the angles between the embeddings. And any solution could be scaled by rescaling $\psi’ =s\psi, \nu’ =\frac{1}{s}\nu$. Normalization would solve this problem.
>
> For $\phi, \mu, u$, although individual motif vectors may not be identifiable, the motif subspace and the induced value function/policy are identifiable. Specifically, the factorization $P(s’|s,a)=\phi(s,a)^\top \mu(s’)$ defines a low-rank transition kernel, and this kernel—and therefore the span of the motifs—is uniquely determined by the data. Any equivalent set of motifs can differ only by an invertible linear transformation (e.g., scaling or rotation), but all such bases generate the same subspace and produce the same value function and policy. This is the same notion of identifiability widely accepted in spectral learning and linear representation methods: the basis is not unique, but the subspace it spans is (a property shared with PCA). In contrast, many latent-variable behavior models (e.g., HMM variants, and autoencoder variants) are non-identifiable even at the subspace level, since different parameterizations can generate exactly the same joint distribution while spanning different latent spaces. Thus, our method enjoys the same structural identifiability guarantees as spectral RL and PCA-like methods: the discovered motif subspace is identifiable, even though the specific basis vectors $\phi$ are not unique up to linear transformation.
>
> **Question 2**
>
> Thanks for the question. Here we wanted to emphasize that their models are based on switching syllables, not continuous changing motifs. The hard boundaries between syllables make switching-syllable-based policy a less natural assumption-the real animal behavior is a continuous mixture of relatively simple patterns. We appreciate that you point out the unclarity and we have changed it to “hidden-Markov-model based” (see the blue text in the introduction).

---

### Official Review · Reviewer_wNqB · 2025-11-01

**Soundness:** 3
**Presentation:** 2
**Contribution:** 2
**Rating:** 4
**Confidence:** 3

**Summary:**

This paper introduces Motif-based Continuous Dynamics (MCD), a novel framework for segmenting and analyzing animal behavior. The method reframes behavior analysis within an imitation learning (RL) context, using spectral decomposition of the transition dynamics to learn a set of task-agnostic "motor motifs." These motifs are then used as a basis to construct continuous, compositional policies that aim to explain observed behavioral trajectories. The authors validate their approach on simulated and real-world datasets, arguing that it offers a more flexible and interpretable alternative to traditional discrete segmentation methods.

**Strengths:**

1.  **Novel Conceptual Framework:** The primary strength of the paper is its innovative application of an RL and imitation learning framework to behavior segmentation. Viewing behavior as the output of a policy optimizing an internal reward is a powerful paradigm shift that promises deeper insights than purely kinematic models.
2.  **Continuous and Compositional Representation:** The method's ability to model behavior as a continuous mixture of motifs is a clear advantage over methods that force a discrete "syllable" at each time point. This naturally handles behavioral transitions and, more importantly, the co-occurrence of different motor programs.

**Weaknesses:**

1. **Missing Architectures:** Key neural network architectures for  $\nu $ and the mapping $f$ in the continuous version are not specified in the main text or the appendix.
    *   **Hyperparameter Sensitivity:** The framework appears to have a large number of hyperparameters (e.g., motif dimensions, learning rates for multiple components, NCE negative samples, GRW priors) that would require significant tuning. The lack of detailed training protocols and ablation studies makes it hard to assess the robustness of the method.

2.  **Fairness of Experimental Comparison:** The quantitative comparison in Section 4.3 using the AUC metric is concerning. MCD's motifs are high-dimensional, continuous functions that explicitly encode the *strength* of each motif's expression ($u(t)$). In contrast, the baseline methods (Keypoint-MoSeq, SemiSeg, OPAL) are primarily designed to output a discrete behavioral label or syllable. By design, MCD's richer, continuous representation has more capacity to fit the nuances of the data. Therefore, outperforming on a likelihood-based metric like AUC seems like an expected outcome of this representational advantage, rather than a clear demonstration of a superior underlying model.

3.  **Limited and Primarily Qualitative Real-World Validation:** While the idea is compelling, the practical advantages on real-world data are not convincingly demonstrated. The analysis is confined to a single freely-moving mouse dataset, and the comparison to state-of-the-art baselines is almost entirely qualitative. While the visualizations are insightful, they are not a substitute for rigorous quantitative comparisons on established benchmarks.

4.  **Lack of Time-Scale Interpretation:** Classical spectral methods on transition matrices provide a natural interpretation of motifs/eigenmodes in terms of the time-scales of the dynamics. It is unclear if the EBM+NCE framework used for the continuous case retains any of this powerful property. The paper does not discuss how the learned motif $\phi$ relate to the long-term or multi-scale structure of the behavior, a feature that methods like MotionMapper explicitly address via wavelet.

**Questions:**

1.  **On Model Architectures and Dimensions:**
    *   **Question 1a:** For Experiment 4.1 (discrete gridworld), you state the number of motifs is 64. Does this mean the motif feature dimension is `d=64`, and the learned `φ` matrix is of size `(9 states * 4 actions) x 64`? Please confirm the relationship between "number of motifs" and the feature dimension `d`.
    *   **Question 1b:** For the continuous version (Sec. 3.2), what are the specific neural network architectures (number of layers, hidden sizes, activation functions) used for `ν(s')` and the mapping `f(ψ)`? This information is essential for reproducibility and is currently missing.

2.  **On Novelty and Relation to Prior Work:**
    *   **Question 2:** How does the motif discovery method for the discrete case differ from the spectral decomposition representation proposed in SPEDER (Ren et al., 2023)? A direct comparison would help clarify the novelty of this part of your contribution.

3.  **On Theoretical Justifications:**
    *   **Question 3a (Time-Scales):** Do the motifs learned via your continuous EBM+NCE framework have an interpretable connection to the time-scales of the behavioral dynamics, analogous to the eigenvalues in classical spectral analysis or the scales in wavelet transforms? If not, does the framework lose the ability to explicitly model multi-scale behavioral organization?
    *   **Question 3b (Single-Step vs. Long-Horizon):** Could you please provide a theoretical argument or an intuitive explanation for why features (`ψ`, `ν`) learned via the single-step NCE objective (Eq. 7) are suitable as a basis for the long-horizon Q-function? Is there an implicit connection that guarantees these locally-optimized features capture global dynamic properties?

4.  **On Experimental Design:**
    *   **Question 4:** Given that MCD uses a continuous, weighted representation while baselines like Keypoint-MoSeq use discrete states, do you agree that a direct comparison on a likelihood-based metric like AUC may inherently favor MCD? Could you propose or discuss an alternative evaluation protocol that might offer a fairer comparison, perhaps by evaluating the quality of the segmentation itself or the downstream utility of the learned representations?

**Minor Comment:**
*   In the paragraph starting at line 36, you state there are "three major limitations" but then proceed to list four distinct points.

---

> ### Author Response · Authors · 2025-11-21
> **Part 1**
>
> **Weaknesses 1**
>
> Thanks for the reply. Some annotations and training protocols were not clear before and now all of them have been clarified in Appendix B, like the details of the network $f$ and $\nu$.
>
> In Appendix J, we have added the discussions of all hyperparameters including motif dimensions, noise distributions, number of negative samples, dataset splitting, GRW prior. Among them, changing the noise distribution to uniform would affect the motif learning quality, and impair performance. Other factors are not so important and could show our model's robustness to hyperparameters.
>
>
> **Weaknesses 2**
>
> Thanks for the reply. We respectfully clarify that the AUC metric does not inherently favor continuous representations over discrete ones. AUC evaluates only a model’s ranking ability—whether positive samples receive higher scores than negative ones—not the dimensionality or continuity of its internal representation. In our setup, every model outputs a scalar score for each state–action pair: Keypoint-MoSeq provides an action log-likelihood from a discrete syllable model; SemiSeg and OPAL provide log-likelihoods based on their continuous latent embeddings; and MCD uses $\phi(s,a)^\top u(t)$ as an unnormalized energy score. Regardless of whether the underlying representation is discrete or continuous, the AUC depends solely on the relative ordering of these scalar scores across positive vs. negative samples. A discrete model can achieve AUC = 1.0 if it consistently assigns higher likelihoods to true actions than mismatched actions, and a continuous model can still perform poorly if its scores are mis-ordered. Thus, AUC reflects ranking quality, not representational capacity. Empirically, we observe that SemiSeg and OPAL—both of which already employ continuous latent representations still underperform compared to MCD. Moreover, when expressing the policy, OPAL parameterizes $\pi(a|s,z(t))$ as a non-linear neural network, while we parameterize $\pi(a|s,t) \propto \exp(\phi(s,a)^\top u(t))$ as an exponential-linear combination. Therefore, they have stronger expressivity and should have performed better than us, but actually didn’t, either quantitatively (Figure 4A) or qualitatively (Figure 4 B,C,D, the curve plots), indicating that MCD’s advantage is not purely representational but reflects better alignment with the underlying pose dynamics.
>
> To further address the reviewer’s concern about the metric fidelity, we additionally evaluated all models using expert labels of the free-moving dataset (Section 4.4), where MCD again outperforms all baselines.
>
> **Weaknesses 3**
>
> First, we have two real-world mouse datasets. The first one is in a discrete labyrinth aiming at testing the navigation ability of the mouse with strong motivations (Section 4.2), and the second one aims at testing the free-moving behavior without goals (Section 4.3). Second, for more rigorous quantitative comparisons, we included expert labels of the animal free-moving behavior dataset and used it to quantitatively evaluate our model in Section 4.4.

---

> ### Author Response · Authors · 2025-11-21
> **Part 2**
>
> **Weaknesses 4**
>
> We would like to clarify that that not all spectral methods yield basis functions with temporal or multi-scale interpretations. There do exist some methods whose motifs/eigenmodes have clear interpretations in terms of time scales. For example, wavelet and Fourier transform use basis vectors of different frequencies to decompose the behaviors. Their basis is temporally distinct as defined. Classical eigendecomposition can also provide such structure. Consider a dynamic system where $P=V\Lambda V^{-1}, x_t=Px_{t-1}=P^tx_0=V\Lambda^t V^{-1}x_0$, then the long-term relationship between $x_t$ and $x_0$ will be dominated by the largest eigenvalue $\lambda_0$ and its corresponding eigenvector $v_0$. However, our method is equivalent to singular value decomposition (SVD, [1]), and SVD basis vectors do not encode distinct temporal frequencies or decay rates. Therefore, the learned motifs $\phi$ are not expected to align with temporal scales in the sense of wavelet, Fourier, or eigenmode decompositions. In fact, our method captures multi-scale behavioral structure through the RL objective, rather than through the spectral basis itself. This is reflected in $u(t)$. Specifically, from the decomposition of state-action value function $Q(s,a,t)=\phi(s,a)^\top u(t)$ where $u(t)=w+\gamma \int V(s’)\mu(s’)q(s’)ds’$, we can see that $u(t)$ integrates two sources of information simultaneously: (i) long-horizon structure through the value function $V(s')$, which summarizes discounted future trajectories, and (ii) short-horizon structure through the immediate reward parameter $w$. Since theoretically the horizon in RL is infinite, our method enables long temporal dependency modeling. Therefore, our method also possesses the long-term and multi-scale temporal modeling capability. Empirically, we observe this structure in Fig.4B: some motifs exhibit stable, slowly varying activation profiles (e.g., motif 1, gray curve), whereas others fluctuate rapidly and capture short-term transitions (e.g., motif 2, pink curve). Intuitively, consider a mouse that intermittently sniffs throughout the day but engages in fast running only during a brief morning period. The coefficient $u_i(t)$ for sniffing would rise gradually across long time windows, while the coefficient $u_j(t)$ for running would increase only during short, specific intervals.
>
> Another difference between our method and MotionMapper is that they capture multi-scale structure in the motion frequency domain, while our method captures this structure in the policy domain from a generative decision process, focusing more on the motivation of the animals.
>
> **Question 1a**
>
> Thank you for the question. Yes—the number of motifs corresponds exactly to the feature dimension $d$. In the discrete setting, the motif matrix $\Phi$ always has shape (n(state)*n(action), $d$). Each column of this matrix is a motif, and each row corresponds to a particular state–action pair. For the gridworld with 9 states and 4 actions, this results in a (36, $d$) matrix. We have changed the description in the main text to clarify that:
> Because the computational complexity only scales linearly with the number of motifs, to cover the motif space as much as possible, we select a large number for motif dimension $d=64$. (See Appendix J for more discussions on the motif dimension $d$.)
>
> **Question 1b**
>
> Thanks. We have added the details about $f$ and $\nu$ in Appendix B.
>
> **Question 2**
>
> We appreciate the reviewer’s request for clarification. Our discrete version indeed builds on the spectral decomposition formulation introduced in SPEDER, and we acknowledge this relationship directly in the paper. For discrete gridworld and labyrinth datasets, we adopt the same low-rank transition factorization because it is theoretically well-grounded and highly suitable for tabular MDPs.
>
> However, the MSE-based motif learning procedure in SPEDER cannot be applied to our more challenging continuous free-moving mouse dataset (See Section 3.2 for more details). To address this, we introduce a new continuous motif learning framework based on energy-based modeling and contrastive estimation, which indirectly learns $\phi$ and scales to high-dimensional continuous trajectories. This part of the pipeline is fundamentally different from SPEDER and constitutes a major methodological innovation in our work.
>
> Importantly, our contribution is not the invention of a new spectral RL estimator per se, but a new behavioral analysis framework grounded in motor motifs and policy decomposition. This relationship is analogous to MotionMapper, whose novelty lies not in the wavelet transform itself, but in how the transform is used to construct a new pipeline for analyzing animal behavior.
>
> [1] Ren, T., Zhang, T., Lee, L., Gonzalez, J. E., Schuurmans, D., & Dai, B. (2022). Spectral decomposition representation for reinforcement learning. arXiv preprint arXiv:2208.09515.

---

> ### Author Response · Authors · 2025-11-21
> **Part 3**
>
> **Question 3a**
>
> We have answered this in Weaknesses 4. The Bellman equation imparts multi-scale modeling capability to the representation $u(t)$, not $\phi(s,a)$. This is different from eigenvalues in eigendecomposition or the basis functions in wavelet transform, whose eigenmodes have temporal structures. Our framework can still model multi-scale behavior dynamics through the flexible $u(t)$.
>
> **Question 3b**
>
> Also see response to Weaknesses 4. Shortly, the long-horizon modeling capability is on $u(t)$, not any of $\psi(s,a)$, $\nu(s’)$, $\phi(s,a)$ or $\mu(s’)$. The Bellman-like equation of u(t) (Eq. 3) imparts long-term modeling capability to it. We also give an intuitive example in that response.
>
> **Question 4**
>
> We have discussed the reasonability of using AUC as a metric to compare MCD and other methods in response to Weaknesses 1, and also included expert labels of the animal free-moving behavior dataset to further support the performance advantage of MCD.

---

> ### Author Response · Authors · 2025-12-04
> **Part 4**
>
> In Appendix J, we have added the discussions of all hyperparameters including motif dimensions, noise distributions, number of negative samples, dataset splitting, GRW prior. Among them, changing the noise distribution to uniform would affect the motif learning quality, and impair performance. Other factors are not so important and could show our model's robustness to hyperparameters. We added additional responses to the "Part 1" comments several days ago. Thank you for taking the time to check them.

---

### Author Response · Authors · 2025-12-04
**We have addressed all concerns proposed by reviewers**

We substantially expanded methodological clarity by specifying all missing architectural details for the continuous model, adding complete training protocols, and providing sensitivity analyses for motif dimension, noise distributions, number of negative samples, and other hyperparameters (Appendix J). We also provided new held-out-subject generalization experiments (Appendix J.3). We clarified the fairness of the AUC comparison by explaining why AUC does not favor continuous representations and we also add expert-label quantitative evaluations, where MCD consistently outperforms all discrete and continuous baselines (Section 4.4). To address concerns about scientific meaningfulness, we expanded the ethology and neuroscience interpretations of motifs (Appendix K). We clarified theoretical questions regarding temporal scales (Appendix I) and identifiability (Appendix H). We systematically positioned the method within RL/IL, robotics-skill literature (Appendix G), and other advanced methods including continuous SSMs (Appendix M). We further added external validation for reward-map experiments (Appendix L) and complementary generative-quality analyses using HMC rollouts (Appendix N). Collectively, these revisions resolve ambiguities, strengthen empirical support, and clearly establish the novelty, robustness, and scientific utility of the proposed motif-based behavioral modeling framework.

---

### Meta-Review · Area_Chair_LMW2 · 2025-12-11

**Summary:**

The reviewers raise many similar concerns. Besides positioning in related work, theoretical justifications, etc,... a central aspect raised by (more or less all) reviewers explicitly is the empirical evaluation of the proposed approach. Reviewers are concerned about the anecdotal and qualitative empirical results and ask for quantitative alternatives. They also raise concerns about the deployed baselines that may be unable to capture fine grained characteristics in comparison to the proposed approach.

**Reviewer Concerns:**

The authors aim to address all concerns of the reviewers. I assume that all or most of the minor questions have successfully been resolved in the rebuttal. HIn response to missing quantitative results, the authors provide a new experiment with manually annotated data (at all 10x20x200 frames, fps unknown) that allows for quantitative evaluation in Section 4.4. Since all reviewers asked for this, I doubt they would be satisfied by only a single new experiment, particularly also because the set of baselines is the same as in the previous experiments. The paper is certainly interesting which is also acknowledged by the reviewers but needs a solid evaluation before publication would be OK.

**Reviewer Scores:**

I believe that all reviewers would acknowledge the explanations and changes made in the paper including also the new experiment but they would at the same time not be willing to change their opinion on the paper because of this one new experiment. The paper is certainly of interest to the community but it needs a much stronger quantitative empirical evaluation before acceptance would be OK. It is important that interested peers can assess the value of the proposed approach for their own research and they necessarily need quantitative results on more than a single data set for doing this.

---

### Decision · Program_Chairs · 2026-01-26

Reject